# Structural mechanisms for VMAT2 inhibition by tetrabenazine

**Michael P Dalton[1], Mary Hongying Cheng[2], Ivet Bahar[2], Jonathan A Coleman[1]***

[1]Department of Structural Biology, University of Pittsburgh, Pittsburgh, United States; [2]Laufer Center for Physical and Quantitative Biology, and Department of Biochemistry and Cell Biology, School of Medicine, Stony Brook University, Stony Brook, United States

**Abstract** The vesicular monoamine transporter 2 (VMAT2) is a proton-dependent antiporter responsible for loading monoamine neurotransmitters into synaptic vesicles. Dysregulation of VMAT2 can lead to several neuropsychiatric disorders including Parkinson's disease and schizophrenia. Furthermore, drugs such as amphetamine and MDMA are known to act on VMAT2, exemplifying its role in the mechanisms of actions for drugs of abuse. Despite VMAT2's importance, there remains a critical lack of mechanistic understanding, largely driven by a lack of structural information. Here, we report a 3.1 Å resolution cryo-electron microscopy (cryo-EM) structure of VMAT2 complexed with tetrabenazine (TBZ), a non-competitive inhibitor used in the treatment of Huntington's chorea. We find TBZ interacts with residues in a central binding site, locking VMAT2 in an occluded conformation and providing a mechanistic basis for non-competitive inhibition. We further identify residues critical for cytosolic and lumenal gating, including a cluster of hydrophobic residues which are involved in a lumenal gating strategy. Our structure also highlights three distinct polar networks that may determine VMAT2 conformational dynamics and play a role in proton transduction. The structure elucidates mechanisms of VMAT2 inhibition and transport, providing insights into VMAT2 architecture, function, and the design of small-molecule therapeutics.

**\*For correspondence:**
coleman1@pitt.edu

**Competing interest:** The authors declare that no competing interests exist.

## eLife assessment

The report presents the cryo-EM structure of human vesicular monoamine transporter 2 (VMAT2) bound to tetrabenazine, a clinical drug. VMAT2 is critical for neurotransmission, and the study constitutes an **important** milestone in neurotransmitter transport research. The evidence presented in the report is **convincing** and provides new opportunities for developing improved therapeutic interventions and furthering our understanding of this vital protein's function.

## Introduction

Neuronal signaling by monoaminergic neurotransmitters controls all aspects of human autonomic functions and behavior, and dysregulation of this leads to many neuropsychiatric diseases. Nearly 60 years ago (*Carlsson et al., 1957*; *Kirshner, 1962*), secretory vesicles prepared from adrenal glands were shown to contain an activity that accumulated epinephrine, norepinephrine, and serotonin in an ATP-dependent manner (*Eiden and Weihe, 2011*). Extensive characterization by many different laboratories of synaptic vesicles (SVs) in neurons showed that monoamine transport activity was also dependent on the proton gradient generated by the V-ATPase, exchanging two protons for one cationic monoamine (*Henry et al., 1998*; *Eiden, 2000*; *Johnson, 1988*; *Eiden et al., 2004*; *Schuldiner et al., 1995*). Monoamine transport was shown to be inhibited by non-competitive inhibitors such as tetrabenazine (TBZ) (*Peter et al., 1994*) and competitive inhibitors like reserpine which has

been used to treat hypertension (*Eiden and Weihe, 2011*). Amphetamines were shown to be mono-amine transporter substrates, eventually leading to vesicle deacidification, disruption of dopamine (DA) packaging in the SV, and DA release into the synapse (*Freyberg et al., 2016*). Cloning of the vesicular monoamine transporter (VMAT) in the 1990s (*Liu et al., 1992*; *Peter et al., 1995*; *Erickson et al., 1992*) revealed two different genes, VMAT1 and VMAT2, that were expressed in the adrenal medulla and brain, respectively (*Peter et al., 1995*; *Weihe et al., 1994*). VMAT2 is expressed in all monoaminergic neurons in the brain, including those for serotonin, norepinephrine, DA, epinephrine, and histamine (*Eiden and Weihe, 2011*), and is essential for loading these neurotransmitters into SVs. VMAT2 is fundamentally required for neurotransmitter recycling and release (*Wang et al., 1997*; *Takahashi et al., 1997*; *Fon et al., 1997*) and changes in VMAT1 and VMAT2 activities either through small-molecule agents or mutations are thought to contribute to many human neuropsychiatric disorders including infantile-onset Parkinson's, schizophrenia, alcoholism, autism, and bipolar depression (*Lohoff et al., 2006*; *Vaht et al., 2016*; *Fehr et al., 2013*; *Bohnen et al., 2006*; *Han et al., 2020*; *Simons et al., 2013*; *Lohoff et al., 2008*).

VMAT1 and -2 are members of the solute carrier 18 (SLC18) family and are also known as SLC18A1 and SLC18A2. The SLC18 subfamily also includes the vesicular transporters for acetylcholine (*Arvidsson et al., 1997*) (VAChT, SLC18A3) and polyamines (*Hiasa et al., 2014*) (VPAT, SLC18B1). Sequence alignments also show that SLC18 transporters belong to the major facilitator superfamily (MFS) of membrane transport proteins which use an alternating access mechanism (*Jardetzky, 1966*; *Mitchell, 1957*) to transport substrate across membranes. SLC18 members are predicted to be comprised of 12 transmembrane (TM) spanning helices (TM1–12), which are arranged in two pseudosymmetric halves each with six TM helices containing a primary binding site for neurotransmitters, polyamines, and inhibitors located approximately halfway across the membrane (*Radestock and Forrest, 2011*; *Figure 1a*). Conformational changes driven by the proton electrochemical gradient are thought to alternatively expose the binding site to either side of the membrane allowing for transport of neurotransmitter from the cytoplasm to the lumen of SVs (*Yaffe et al., 2013*; *Yaffe et al., 2016*).

VMAT2 and VMAT1 share 62% sequence identity but have distinct substrate specificity and pharmacological properties (*Erickson et al., 1996*). Small-molecule ligands such as TBZ and reserpine are high-affinity inhibitors of VMAT2, which prevent neurotransmitters from binding, arrest VMAT2 from cycling, and consequently reduce neuronal signaling. VMAT2 exhibits higher affinity for TBZ, monoamines, and amphetamines, whereas reserpine binds equally to both VMAT2 and VMAT1 (*Peter et al., 1994*). TBZ is the only drug which is approved for treatment of chorea associated with Huntington's disease and has shown to be effective in various other hyperkinetic conditions such as tardive dyskinesia, dystonia, tics, and Tourette's syndrome (*Kaur et al., 2016*). A proposed mechanism for TBZ inhibition of VMAT2 involves two sequential steps, initial low-affinity binding of TBZ to the lumenal-open state of VMAT2 which produces a conformational change, resulting in a high-affinity dead-end TBZ-bound occluded complex (; *Scherman et al., 1983*; *Scherman and Henry, 1984*; *Ugolev et al., 2013*).

Here, we report a structure of VMAT2 bound to TBZ at 3.1 Å resolution in an occluded conformation using single-particle cryo-electron microscopy (cryo-EM), describing the architecture of VMAT2, identifying the high-affinity TBZ binding site, and revealing the mechanisms of drug and neurotransmitter binding, inhibition, and transport.

## Results

### Cryo-EM imaging of human VMAT2

Since VMAT2 is a small monomeric membrane protein of approximately 55 kDa, cryo-EM structure determination is challenging. To overcome this, we incorporated mVenus and the anti-GFP nanobody into the N- and C-terminus respectively of human VMAT2 to provide mass and molecular features to facilitate the cryo-EM reconstruction (*Figure 1b*), this created a hook-like fiducial feature by reconstituting the interaction of these proteins on the cytosolic side of VMAT2 (*Kubala et al., 2010*). Attachment of both proteins to the termini proved to be ineffective as the unstructured N- and C-terminus of VMAT2 are flexible; to combat this problem, we determined the minimal termini length that would reduce flexibility while maintaining VMAT2 folding. After successive optimizations, our final construct contained mVenus fused to the N-termini at position 17, and the anti-GFP nanobody at position 482

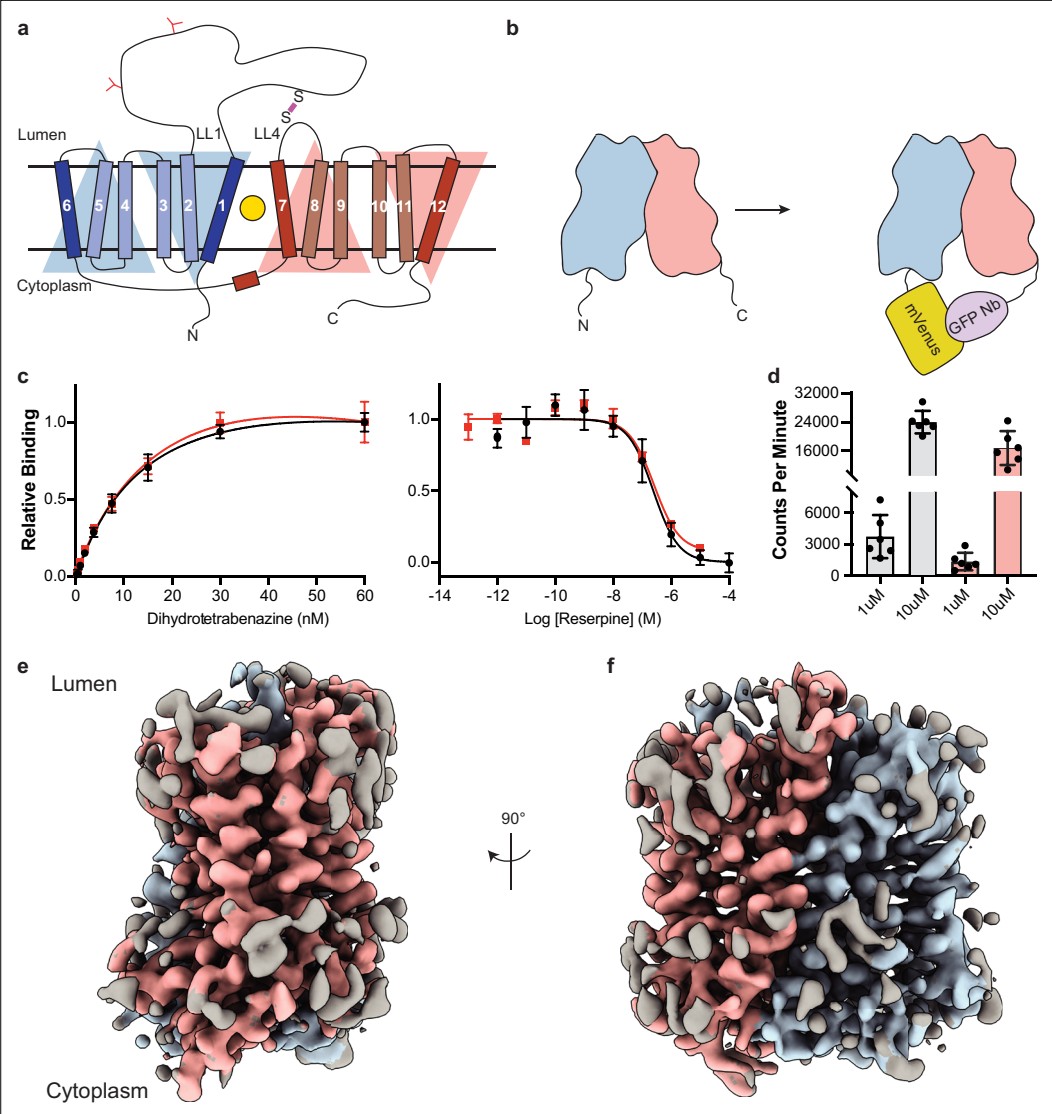

**Figure 1.** Cryo-electron microscopy (cryo-EM) reconstruction and functional characterization of vesicular monoamine transporter 2 (VMAT2)-tetrabenazine complex. (**a**) Predicted structural elements of VMAT2. The neurotransmitter substrate is bound at the central site (yellow, circle) between the two repeats comprised of transmembrane (TM)1–6 and 7–12. The red and blue triangles depict the pseudo twofold symmetric repeats. A disulfide bond (purple line) is predicted between lumenal loops LL1 and LL4. The N-linked glycosylation sites in LL1 are shown as red 'Y' shapes. (**b**) Intrinsic fiducial strategy involves attachment of mVenus and GFP-Nb to the N- and C-terminus of VMAT2. (**c**) Left panel, plots of [³H]-DTBZ saturation binding to wild-type VMAT2 (black, circles) and chimera (red, squares). Symbols show the mean derived from n=3 technical replicates. Error bars show the s.e.m. Right panel, graphs of competition binding of ³H-DTBZ with unlabeled reserpine, error bars show the s.e.m. DTBZ, dihydrotetrabenazine. (**d**) Plots of transport into vesicles using 1 and 10 µM ³H-serotonin for wild-type VMAT2 (gray bars) and chimera (red bars). The bars show the means and points show the value for each technical replicate. Error bars show the s.e.m. (**e, f**) Occluded map of VMAT2-tetrabenazine complex (3.1 Å resolution, contour level 0.336). The mVenus and GFP-Nb fiducial is not shown for clarity.

The online version of this article includes the following figure supplement(s) for figure 1:

**Figure supplement 1.** Biochemical characterization, construct design, and sequence conservation of vesicular monoamine transporter 2 (VMAT2).

**Figure supplement 2.** Cryo-electron microscopy (cryo-EM) data processing of the vesicular monoamine transporter 2 (VMAT2)-tetrabenazine complex.

**Figure supplement 3.** Cryo-electron microscopy (Cryo-EM) maps and interpretation of vesicular monoamine transporter 2 (VMAT2) reconstruction.

which we denote the VMAT2 chimera (*Figure 1—figure supplement 1a–e*). We investigated the consequences of VMAT2 modification to ensure the chimera maintained functional activity. First, we performed binding experiments with $^3$H-labeled dihydrotetrabenazine (DTBZ) and found the chimera-bound DTBZ with a similar affinity ($K_d$ = 26 ± 9 nM) to the wild-type control ($K_d$ = 18 ± 4 nM) (*Figure 1c*). Competition binding of labeled DTBZ with unlabeled reserpine stabilizes cytoplasmic-open (*Yaffe et al., 2018*), a state which is incompatible with TBZ binding and produced a $K_i$ of 173±1 nM for reserpine, which was like wild type (161±1 nM). Next, we performed transport experiments using permeabilized cells, initial time course experiments using $^3$H-labeled serotonin showed clear accumulation (*Figure 1—figure supplement 1f*), and steady-state experiments using 1 and 10 µM serotonin measured within the linear uptake range showed similar transport activity as wild-type VMAT2 (*Figure 1d*). Thus, the functional properties of the chimera are comparable to wild-type VMAT2.

To understand the architecture, locate the drug binding site, and assess how TBZ binding influences the conformation of the transporter, we studied the VMAT2 chimera using single-particle cryo-EM (*Figure 1e and f*, *Figure 1—figure supplement 2*). The resulting cryo-EM map was determined to a resolution of 3.1 Å, the densities of the TM helices were well defined, continuous, and exhibited density features for TBZ in the primary binding site and most of the side chains (*Table 1*, *Figure 1—figure supplement 3*). This demonstrates the feasibility of our approach and enabled us to build a model of VMAT2.

## Architecture of VMAT2

The TBZ-bound VMAT2 complex adopts an occluded conformation, with TBZ in a binding site between the central TM helices. The 12 TM helices of the transmembrane domain (TMD) of VMAT2 are arranged in a tight bundle with TM1–6 and TM7–12 each organized into a pseudosymmetrical half (*Figure 2a*). The cytosolic facing side of VMAT2 is characterized primarily by the unstructured N- and C- termini along with a 20-residue loop that connects the two halves, extending from TM6 to TM7 before terminating in a short α-helix that runs parallel to the bilayer and connects to TM7 with a short linker. TM5 and -11 both contain proline residues near the lumenal face, which break the helical structure and facilitate connections to TM6 and -12 respectively. TM9 and -12 exhibit significant heterogeneity in our cryo-EM reconstructions; we speculate that this is likely due to a dynamic nature intrinsic to the TMs, an aspect that may offer a glimpse into VMAT2 dynamics.

**Table 1.** Cryo-electron microscopy (cryo-EM) data collection, refinement, and validation statistics.

| | VMAT2-TBZ (EMDB-41269) (PDB 8THR) |
|---|---|
| **Data collection and processing** | |
| Magnification | 77,279 |
| Voltage (kV) | 300 |
| Electron exposure (e⁻/Å$^2$) | 60 |
| Defocus range (µm) | –1.0 to –2.5 |
| Pixel size (Å) | 0.647 |
| Symmetry imposed | C1 |
| Initial particle images (no.) | ~10,000,000 |
| Final particle images (no.) | 65516 |
| Map resolution (Å) FSC threshold | 3.12 0.143 |
| Map resolution range (Å)* | 6.6–2.8 |
| **Refinement** | |
| Initial model used (PDB code) | |
| Model resolution (Å) FSC threshold | 3.7 0.5 |
| Model resolution range (Å) | 20.4–3.7 |
| Map sharpening B-factor Å$^2$ | |
| Model composition Non-hydrogen atoms Protein residues Ligands | 2937 389 1 |
| B-factors Å$^2$ Protein Ligand | 28.99 55.22 |
| R.m.s. deviations Bond lengths (Å) Bond angles (°) | 0.0004 (0) 0.37 (15) |
| Validation MolProbity score Clashscore Poor rotamers (%) | 1.23 4.34 0.96 |
| Ramachandran plot Favored (%) Allowed (%) Disallowed (%) | 98.0 2.00 0.00 |

*Local resolution range at 0.5 FSC.

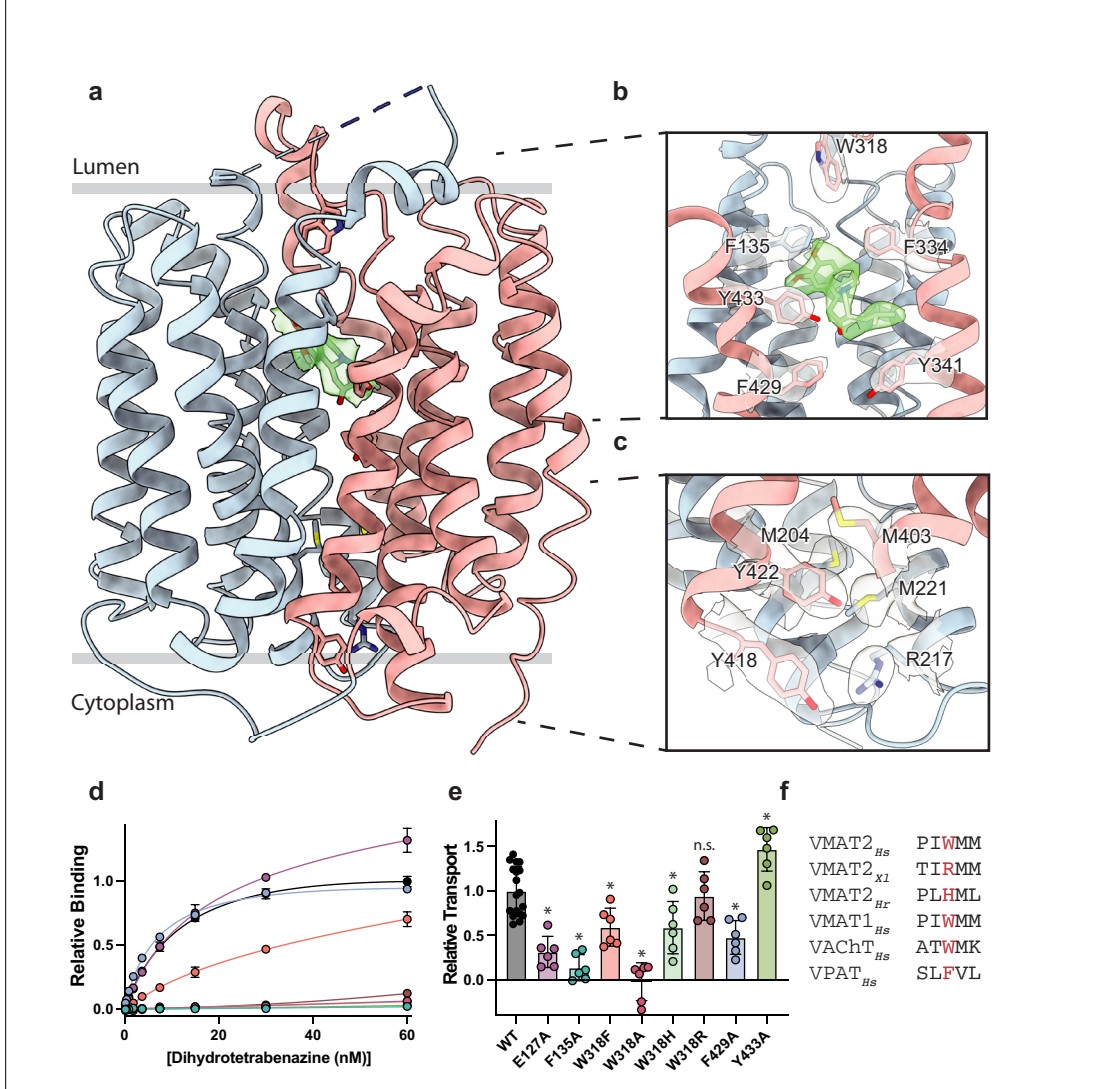

**Figure 2.** Vesicular monoamine transporter 2 (VMAT2) conformation and residues involved in gating. (**a**) Overall view of the VMAT2-tetrabenazine (TBZ) complex. TBZ is shown in light green sticks with its map density in transparent surface. (**b**) Closeup view of lumenal gating residues and TBZ, shown in stick representation together with transparent surface representation of their map density. (**c**) Cytosolic gating residues, same representation as in (**b**). (**d**) Binding of dihydrotetrabenazine (DTBZ) to various VMAT2 mutants in the lumenal and cytosolic gates including wild type (black), W318R (brown), W318H (light green), E127A (purple), W318F (orange), Y433A (forest green), F429A (blue), W318A (red), and F135A (teal). Data were normalized to wild type with error bars denoting standard deviation. (**e**) Serotonin transport activity of lumenal and cytosolic gating residue mutants. Symbols show the mean derived from n=6 technical replicates with an identical color scheme to (**d**). Asterisks denote statistical significance from wild type, with no significance being denoted with n.s. Data were normalized to wild-type transport. Statistics were calculated in GraphPad Prism using a one-way ANOVA with Dunnett's multiple comparison test. Error bars show the standard deviation. (**f**) Alignment of five sequential residues of human VMAT2 (two residues on either side of W318) against their counterparts in *Xenopus laevis* (*Xl*), *Helobdella robusta* (*Hr*), and VMAT1, VAChT, and VPAT from humans. The residues which align with W318 in VMAT2 are shown in red.

The online version of this article includes the following figure supplement(s) for figure 2:

**Figure supplement 1.** Sequence alignment of vesicular monoamine transporter 1 (VMAT1), VMAT2, VAChT, and VPAT.

VMAT1 and -2 encode a large lumenal loop (LL) 1 which contains several N-linked glycosylation sites (*Yao and Hersh, 2007*) and a disulfide bridge between LL1 and LL4 (*Thiriot et al., 2002*). LL1 and -4 also contain intriguing elements of structure: LL1 extends into the lumenal space in an unstructured loop which is mostly not resolved in our structure, before terminating in a short helix which interacts with the lumenal face of the transporter near TM7, -11, and -12; LL4 extends outward from TM7 into the lumen before connecting back to TM8. A striking feature of LL4 is the location of W318

which positions its indole side chain directly into a lumenal cavity near the TBZ site, acting as a plug to completely occlude the lumenal side of the transporter (*Figure 2b*). Together, these loops cinch the lumenal side of the transporter closed, locking VMAT2 in an occluded conformation and preventing ligand egress. The conserved nature of LL4 and W318 suggests this motif is necessary for transport function and is a key player in the transport mechanism (*Figure 1—figure supplement 1b and g*, *Figure 2—figure supplement 1*). The conformation of LL1 and -4 is also likely aided by a disulfide bond between cysteines 117 and 324, which is known to be necessary for transporter function (*Thiriot et al., 2002*), our structure was not able to unequivocally place this bond due to the lack of density for residues 48–118 of LL1.

## Cytosolic and lumenal gates

The structure of the VMAT2-TBZ complex reveals that both the cytosolic and lumenal gates are closed, which precludes solvent and ligand access from either the cytosolic or lumenal compartments (*Figure 2a–c*). Previous studies have suggested that residues R217, M221, Y418, and Y422 make up the cytosolic gate (*Yaffe et al., 2016*). We find R217 and Y418 form the outer cytosolic gate with the guanidino group of R217 involved in a cation-π interaction with the aromatic benzyl group of Y418 which seals off cytosolic access to the binding site (*Figure 2c*). M221 and Y422 form a second set of cytosolic gating residues 'above' the outer cytoplasmic gate through a stable methionine-aromatic interaction which acts to fully seal the cytoplasmic gate (*Figure 2c*). It is likely that M204 and M403 also contribute to cytosolic gating in this region as their side chains also act to fill this space. On the lumenal side, F135, F334, and W318 form the lumenal gate where they interact with one another to block access to the binding site. W318 acts as 'cap' with the indole side chain facing into a tightly packed hydrophobic pocket consisting of residues I44, V131, L134, L315, I317, and I381 which completely prevent access on the lumenal side. W318 is highly conserved in the SLC18 family, suggesting that SLC18 transporters share a common mechanism of lumenal-gate closure (*Figure 2—figure supplement 1*). E127 of LL1 may play a role in stabilizing the tryptophan in this conformation, with the carboxyl group of the side chain orienting itself near the indole nitrogen potentially forming a hydrogen bond pair. We found that mutation of this residue to alanine did not significantly reduce TBZ binding relative to wild type (*Figure 2d*). The inner gate is located just below TBZ and comprises residues Y341, F429, and Y433 (*Figure 2b*; *Yaffe et al., 2016*). When probed for their role in inhibitor

**Table 2.** Calculated $K_d$ values of dihydrotetrabenazine (DTBZ) for various single-point mutants.

| Mutant | $K_d$ (nM) |
|--------|-----------|
| WT | 15±3 |
| N34A | ND |
| N34D | ND |
| N34Q | ND |
| N34T | ND |
| E127A | 15±6 |
| F135A | ND |
| R189A | ND |
| V232L | ND |
| E312Q | ND |
| W318A | ND |
| W318F | 24±16 |
| W318H | ND |
| W318R | ND |
| F429A | 7.7±0.6 |
| Y433A | ND |

binding, we found that alanine mutants of F135, Y433, and W318 all greatly reduced DTBZ binding (*Figure 2d*). Extensive contacts of TBZ with F135 may function to keep the transporter closed on the lumenal side, which would trap VMAT2 in the occluded conformation. F135 and Y433 form π-stacking interactions with TBZ which coordinate the benzene ring of TBZ. F429A did not reduce DTBZ affinity ($K_d$ = 7.7 ± 0.6 nM) compared to the wild-type control ($K_d$ = 15 ± 2 nM), showing that while mutation of this residue compromises the inner cytosolic gate (*Figure 2b*), it is not directly involved in binding TBZ. Conversely, while W318 in LL4 also does not interact directly with TBZ, W318 is required for stabilizing the occluded conformation, and replacement with alanine prevents TBZ from being trapped inside the transporter by preventing closure of the lumenal gate. Sequence alignment with other members of the SLC18 family reveals broad conservation of W318 except for VAChT which contains a phenylalanine in its place (*Figure 2—figure supplement 1*) and a W318F mutation retained some DTBZ binding ($K_d$ = 24 ± 16 nM) (*Table 2*, *Figure 2d*). Alignment of VMAT2 sequences from other species also shows a high degree of conservation of W318 with some notable exceptions which substitute W318 for a large positively charged residue such as an arginine or histidine; and upon investigation we found these mutants all greatly diminish DTBZ binding (*Figure 2d*).

To further investigate the functional role of the identified gating residues, we performed serotonin transport experiments. We found that mutation of both E127 and F135 to alanine significantly reduced transport activity (*Figure 2e*). E127A produced a large reduction in transport, resulting in just 34% activity. Replacing F429 with alanine reduced transport as well, but only to about half the level of wild type (*Figure 2e*). Interestingly, the Y433A mutation appeared to enhance transport, and while critical for TBZ binding, this mutant does not prevent transporter cycling (*Figure 2d and e*). Mutation of W318 to alanine greatly reduced transport, paralleling the effect observed with DTBZ binding

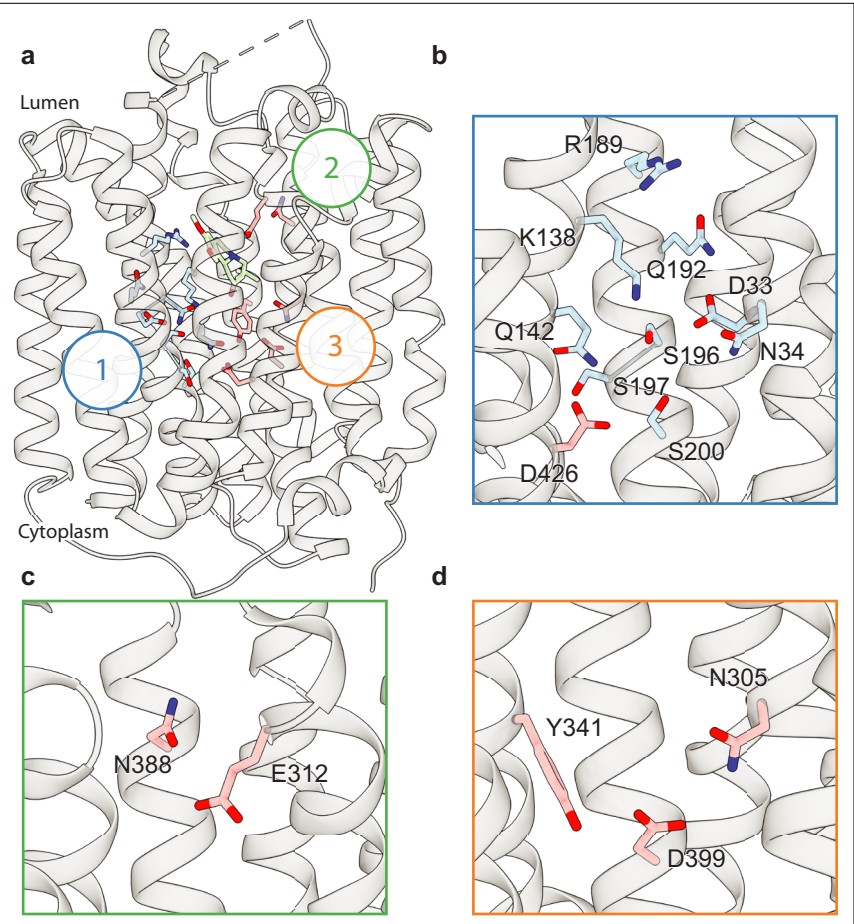

**Figure 3.** Polar networks. (**a**) Overall view showing three distinct polar networks. Polar residues involved in each network and tetrabenazine (TBZ) are shown in sticks. (**b**) Cartoon representation showing a zoomed view of polar network 1. (**c**) Polar network 2. (**d**) Polar network 3.

(*Figure 2d and e*) while a histidine mutant at this position maintained a significant amount of transport activity (*Figure 2d–f*) and the phenylalanine substitution had about half the activity of wild type. The highest activity of the examined W318 mutants was W318R, which fully recapitulated the transport activity of wild type despite being unable to bind DTBZ (*Figure 2d and e*).

## Polar networks

Upon careful inspection of the model, we were able to identify distinct polar networks that we believe may play a role in proton coordination and subsequent transporter conformational change (*Figure 3a*). The first and largest of these networks lies between TM-1, -4, and -11, and consists of residues D33, N34, K138, Q142, R189, Q192, S196, S197, S200, and D426 (*Figure 3b*; *Yaffe et al., 2013*). At the center of this network lies D33 (*Yaffe et al., 2013*), which makes critical contacts with the side chains of N34, K138, S196, and Q192. Together, the residues comprise a complex hydrogen bond network linking TM1, -4, and -11. D426 (*Merickel et al., 1997*) lies further toward the cytosol with the side chain carboxyl group facing the bulk of the network, likely forming a hydrogen bond with the hydroxyl group of S200. In the other TMD half there are two distinct groups of interacting polar residues, which bridge between TM7, -8, and -10 (*Figure 3a, c, and d*). The second group is a pair of residues found on the lumenal side, between residues E312 and N388 with the amide group of the N388 side chain pointed toward the carboxyl group of E312, which could act to stabilize TBZ in the binding site (*Figure 3c*). The third group is located toward the cytosolic side and consists of N305, Y341, and D399 (*Merickel et al., 1997*), the latter two of which have previously been speculated to form a hydrogen bond pair (*Yaffe et al., 2013*). The side chains of these residues are positioned toward one another, with the carboxyl group of D399 forming a hydrogen bond with N305 and likely Y314 (*Figure 3d*).

## TBZ binding site

The resolution of our map allowed us to unambiguously place TBZ in the central binding site (*Figure 4a and b*). TBZ adopts a pose which is predominantly perpendicular to the direction of the TM helices in the lumenal half of VMAT2 near the location of the lumenal gating residues. The TBZ binding site exhibits an amphipathic environment, comprised of both polar and non-polar residues where the tertiary amine of TBZ orients itself toward the negatively charged surface of the binding site near TM7 and -11, and toward E312 (*Figure 4c*, *Figure 4—figure supplement 1a*). We considered that E312 may play an analogous role to the highly conserved aspartate residue present in neurotransmitter sodium symporters such as the serotonin, DA, and norepinephrine transporters (*Yamashita et al., 2005*) which utilize a negatively charged residue to directly bind to amine groups (*Figure 4d and e*, *Figure 2—figure supplement 1*). We thus performed radiolabeled binding experiments to assess the effect of mutating residues in the TBZ binding site by measuring binding of $^3$H-labeled DTBZ (*Figure 4f*, *Table 2*). E312 was previously shown to be necessary for substrate transport and inhibitor binding, so we first selected this residue for mutagenesis to probe its importance in TBZ binding (*Yaffe et al., 2013*; *Støve et al., 2022*). The E312Q mutant did not fully abolish DTBZ binding (*Figure 4—figure supplement 1b*) but did greatly reduce DTBZ affinity. This demonstrates that, while not completely essential, E312 is important for inhibitor binding and likely substrates transport by interacting with the amine of the neurotransmitter (*Figure 4d and e*). Next, we observed that R189 orients its guanidino group toward the methoxy groups of TBZ likely forming hydrogen-bonding interactions and we found that replacement of R189 with an alanine nearly completely abolished DBTZ binding at all concentrations tested (*Figure 4f*, *Figure 4—figure supplement 1b*). The high degree of conservation of this residue suggests that it plays an important role in transporter function, and even a conservative substitution to a lysine nearly eliminated DTBZ binding (*Figure 4f*, *Table 2*). K138 has been previously shown to play an important role in both TBZ binding and serotonin transport and the primary amine side chain is positioned toward the TBZ binding site (*Merickel et al., 1997*; *Figure 4d and e*). K138 is positioned between two aspartate residues (D426 and D33) and is part of a hydrogen bond network that has been previously hypothesized (*Merickel et al., 1997*). Previous experiments found that mutating K138 to alanine resulted in an approximate fourfold reduction in TBZ binding affinity (*Yaffe et al., 2013*) but did not extinguish TBZ binding and therefore K138 is likely involved in direct interactions with substrate or inducing conformational changes during proton transport and is not directly involved in TBZ binding. N34 is of particular interest since the amide group of its side chain appears to form a hydrogen bond with the carbonyl oxygen of TBZ (*Figure 4d and e*) which

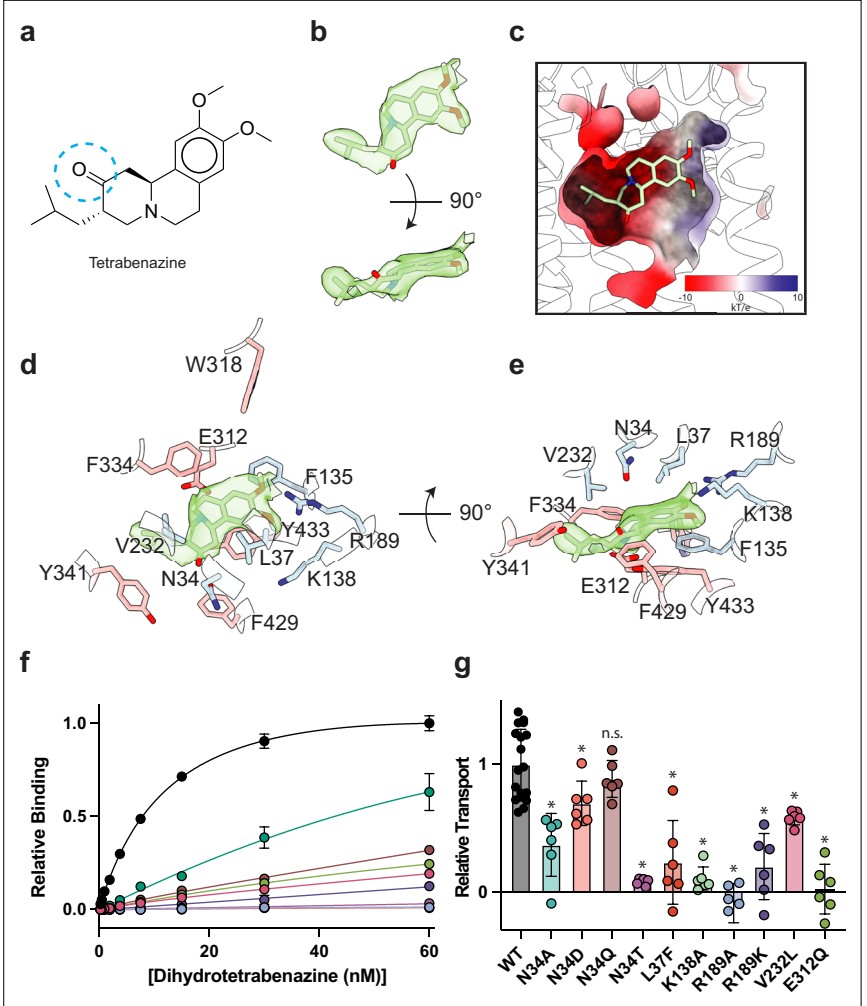

**Figure 4.** Tetrabenazine (TBZ) recognition and binding. (**a**) Chemical structure of TBZ. The blue dotted circle indicates the position of the hydroxyl group in dihydrotetrabenazine (DTBZ). (**b**) Density associated with TBZ is shown in green transparent surface sharpened with a B-factor of –50 Å². TBZ is shown in sticks fit to the density. (**c**) Electrostatic potential of the TBZ binding site. (**d, e**) Binding site of TBZ, residues which are involved in binding are shown in sticks. Density associated with TBZ is shown in light green surface. (**f**) Plots of ³H-DTBZ saturation binding to wild type (black), V232L (pink), R189A (blue), E312Q (forest green), N34D (orange), N34T (light purple), N34A (teal), N34Q (brown) R189K (purple) L37F (red), and K138A (light green). (**g**) Serotonin transport activity of mutants in TBZ binding site. Symbols show the mean derived from n=6 technical replicates with an identical color scheme to f. Asterisks denote statistical significance from wild type, with no significance being denoted with n.s. Statistics were calculated in GraphPad Prism using a one-way ANOVA with Dunnett's multiple comparison test. Error bars show the s.e.m.

The online version of this article includes the following video and figure supplement(s) for figure 4:

**Figure supplement 1.** Point mutants in tetrabenazine (TBZ) binding site.

**Figure supplement 2.** Tetrabenazine (TBZ) docking and molecular dynamics (MD) simulations.

**Figure supplement 3.** Effects of protonation states of tetrabenazine (TBZ) and selected vesicular monoamine transporter 2 (VMAT2) residues on the time evolution of TBZ (heavy atoms) and VMAT2 (α-carbons) root-mean-square-deviations (RMSDs) from their original (cryo-electron microscopy [cryo-EM]-resolved) positions.

**Figure 4—video 1.** Binding and coordination of neutral tetrabenazine (TBZ) (cyan van der Waals [vdW] representation) observed in a 180 ns molecular dynamics (MD) simulation (run 1). https://elifesciences.org/articles/91973/figures#fig4video1

**Figure 4—video 2.** Binding and coordination of neutral tetrabenazine (TBZ) (cyan van der Waals [vdW] representation) observed in a 180 ns molecular dynamics (MD) simulation (TBZ_1 run 2), in the same format as

*Figure 4 continued on next page*

*Figure 4 continued*

***Figure 4—video 1***.
https://elifesciences.org/articles/91973/figures#fig4video2

**Figure 4—video 3.** Binding and coordination of neutral tetrabenazine (TBZ) (cyan van der Waals [vdW] format) observed in a 180 ns molecular dynamics (MD) simulation (TBZ_1 run 3), in the same format as *Figure 4—video 1*.
https://elifesciences.org/articles/91973/figures#fig4video3

is a hydroxyl group in DTBZ. DTBZ binding was not detectable to N34 mutants of either glutamine, threonine, or aspartate while substitution to alanine preserved some binding (*Figure 4f*, *Table 2*, *Figure 4—figure supplement 1b*).

To investigate the functional role of residues in the binding site, we again performed a series of serotonin transport experiments. We found R189A, E312Q, N34T, N34A, and K138A mutations all exhibited reduced transport activity (*Figure 4g*). R189A and E312Q exhibited the largest change, reducing transport to essentially zero. Substitution of N34 with glutamine had little or no effect on transport activity, opposite of what was noted for DTBZ binding. Replacing N34 with alanine was detrimental, reducing transport to less than half of wild type (*Figure 4g*). We found N34D and N34T to have opposing effects, with N34D having activity slightly less than wild type and N34T having little to no transport activity at all. R189K greatly affected transport but retained some activity despite lacking measurable DTBZ binding.

Our model of the VMAT2-TBZ complex allowed us to pinpoint two residues which contribute to the specificity of TBZ to VMAT2 over VMAT1. Previous studies have highlighted V232 (*Støve et al., 2022*), which is a leucine in VMAT1, as being putatively involved in conferring differences in affinity, and our model shows that V232 is positioned closely to the isobutyl of TBZ which is wedged into a small hydrophobic pocket (*Figure 4—figure supplement 1*). The addition of an extra carbon of the leucine side chain would produce a steric clash and limit the ability of TBZ to bind (*Figure 4e*). The V232L mutant in VMAT2 reduces the affinity of DTBZ to VMAT2, confirming its importance in specificity, but the V232L mutant did not show a complete loss in binding (*Figure 4f*, *Figure 4—figure supplement 1b*). Therefore, we carefully inspected the binding site of VMAT2 and compared it to the predicted structure of VMAT1 to find additional differences in the binding site (*Figure 2—figure supplement 1*), we found that L37 in VMAT2 is a phenylalanine in VMAT1 (*Figure 4—figure supplement 1c*). Given its proximity

to TBZ, this substitution would produce a steric clash with the benzene ring (*Figure 4—figure supplement 1a and c*). We found that the L37F mutant resulted in nearly no detectable binding of DTBZ at 60 nM concentration (Figure 4-figure supplement 4b). Thus, the combination of these two substitutions likely constitutes the differences in TBZ affinity of VMAT2 vs. VMAT1. Interestingly, V232L and L37F both retained some transport activity with L37F producing a more significant reduction in activity (*Figure 4g*).

Our docking simulations suggested that TBZ may sample two different binding poses by small reorientation and movements within the same binding pocket (*Figure 4—figure supplement 2*), both exhibiting similar binding affinities (–9.5±0.2 kcal/mol); the first pose is almost identical to that resolved in our cryo-EM structure (0.4 Å root-mean-square-deviation [RMSD], *Figure 4—figure supplement 2e*); and the second (3 Å RMSD in TBZ heavy atom coordinates) shows a reorientation of the TBZ methoxy groups toward C430, a residue previously identified to play an important role in binding TBZ (*Thiriot and Ruoho, 2001*; *Figure 4—figure supplement 2f*). These two

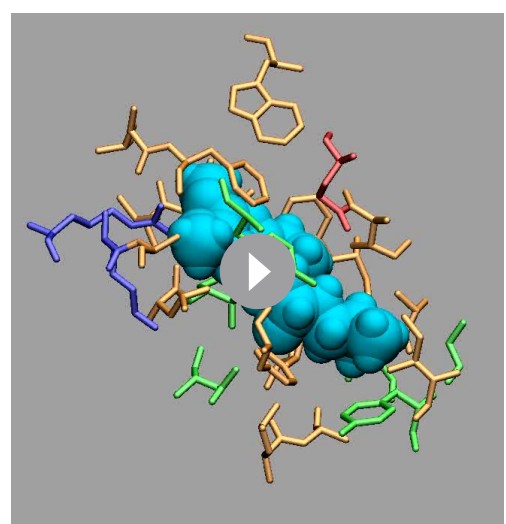

**Video 1.** Same as *Figure 4—videos 1–3*, with neutral tetrabenazine (TBZ), but with the additional protonation of D426 (TBZ_3 run 1).(Related to *Table 3*). TBZ adopted the predominant pose similar to that resolved in the cryo-electron microscopy (cryo-EM) structure.

https://elifesciences.org/articles/91973/figures#video1

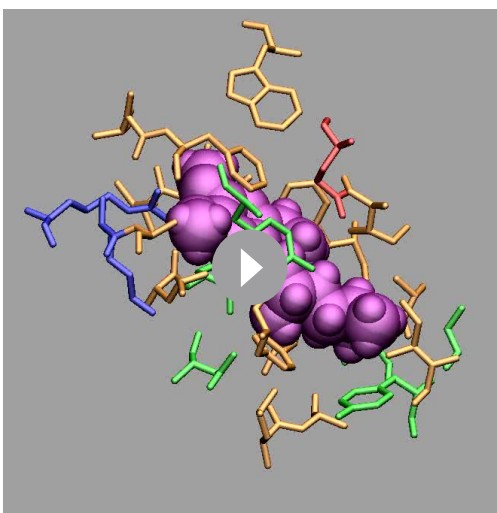

**Video 2.** Molecular dynamics (MD) simulation of the binding and coordination of protonated tetrabenazine (TBZ) (TBZ⁺; *pink* van der Waals [vdW] format) to vesicular monoamine transporter 2 (VMAT2) with protonated E312 and D399. (Related to *Table 3*). This is a 100 ns run, termed TBZ_2 run 1. The same format as *Figure 4—videos 1–3* is adopted for the coordinating residues. TBZ tends to alter its binding pose to approximate the one observed in *Figure 4—video 2*.
https://elifesciences.org/articles/91973/figures#video2

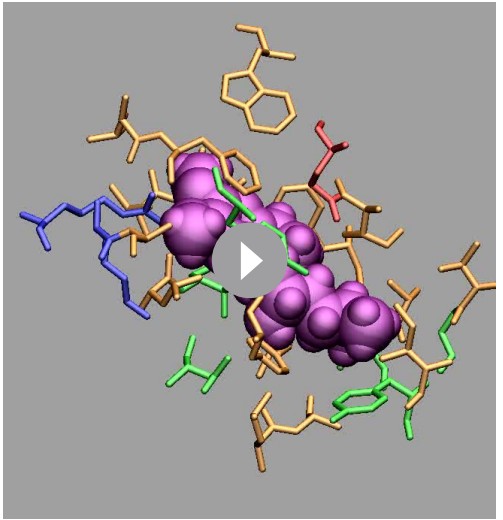

**Video 3.** Same as *Video 1*, except for the protonation of tetrabenazine (TBZ) (TBZ⁺ shown in pink van der Waals [vdW] representation). (Related to *Table 3*).This is a 100 ns run, termed TBZ_4 run 1. In the presence of protonation, TBZ preferentially samples the pose observed in *Figure 4—video 2*, *Video 2*.
https://elifesciences.org/articles/91973/figures#video3

poses were also observed in molecular dynamics (MD) simulations as illustrated in *Figure 4—figure supplement 2a–d* and *Figure 4—videos 1–3*, *Video 1*. The second pose provides insights into the adaptability of TBZ to the conformational dynamics of VMAT2, while it preferentially positions itself into the pocket resolved in our cryo-EM map. TBZ is thought to enter from the lumenal side of VMAT2 by binding to the lumenal-open conformation (*Ugolev et al., 2013*). It may interact first with C430 and the other coordinating residues at this pose before R189 moves between the two methoxy groups and allows TBZ to settle into the resolved orientation (*Figure 4—figure supplement 2f*). This result highlights the stepwise events that inhibitors like TBZ may undergo to stably bind to their targets.

The binding stability of TBZ is also influenced by its protonation state. When TBZ is protonated (TBZ⁺), it induces the diffusion of three to four times more water molecules within the TBZ binding pocket compared to neutral TBZ (*Table 3*, *Figure 4—figure supplement 3*). This flux of water results in the dissociation of TBZ from its binding site as illustrated in *Videos 2 and 3*. Several titratable residues, including D33, E312, D399, D426, K138, and R189, line the central cavity of VMAT2 and impact TBZ binding stability (*Table 4*). We found that maintaining an overall neutral charge within the TBZ binding pocket, as observed in system TBZ_1, most effectively preserves the TBZ-bound occluded state of VMAT2. Residues R189 and E312 in particular are within close proximity of TBZ and participate directly in binding.

## Neurotransmitter release

To examine the binding propensity of DA to VMAT2 in the occluded conformation, we constructed five simulation systems with varying protonation states for the four acidic residues (D33, E312, D399, and D426) that line the binding pocket (*Table 5*). For all five systems, DA carried a +1 charge and was initially placed with a pose predicted by docking simulations to be the most favorable binding pose (*Figure 5—figure supplement 1a and b*); and for each system, two MD runs of 100 ns were performed, except for the case where all acidic residues were in a protonated state of which one of the runs was extended to 200 ns to visualize the release of DA to the vesicular lumen. The simulations revealed alterations in DA's binding properties depending on the protonation states of the four acidic

**Table 3.** Molecular dynamics (MD) simulation systems of vesicular monoamine transporter 2 (VMAT2) in the presence of tetrabenazine (TBZ), their properties and simulation durations.

| System ID | Protonated residues | Bound ligand | Duration (ns) | Waters in binding pocket* | RMSD in VMAT2 Cα-atoms (Å)† | RMSD in TBZ heavy atoms (Å) ‡ |
|---|---|---|---|---|---|---|
| TBZ_1 | | TBZ | 3×180 | 2.8±1.3 | 2.4±0.3 | 2.7±0.5 |
| TBZ_2 | E312 and D399 | TBZ⁺ | 2×100 | 8.1±1.4 | 2.4±0.4 | 5.9±1.4 |
| TBZ_3 | E312, D399, and D426 | TBZ | 3×100 | 4.6±1.4 | 2.7±0.5 | 3.5±1.3 |
| TBZ_4 | | TBZ⁺ | 2×100 | 14.3±2.4 | 2.3±0.2 | 3.7±0.4 |

*Number of water molecules within 3.5 Å of TBZ (or TBZ⁺) averaged over multiple MD snapshots.
†RMSDs (root-mean-square-deviations) of VMAT2 Cα-atoms from their initial (cryo-EM resolved) positions.
‡RMSD of the heavy atoms of TBZ from their cryo-EM-resolved positions. All averages and standard deviations were calculated between 50 and 100 ns portion of the MD trajectories.

residues (*Figure 5—figure supplement 1c*, *Table 5*). Two notable differences were observed when comparing DA to TBZ binding (system DA_2 vs. TBZ_1). First, upon binding TBZ, R189 orients its guanidino group toward the methoxy groups of TBZ and forms hydrogen bonds in both the cryo-EM structure and MD simulations; in the case of DA, hydrogen bond formation of the hydroxyl groups

**Table 4.** p$K_a$ calculations performed by PROPKA 3.5.0.

| Residue | p$K_a$ with TBZ | p$K_a$ without TBZ | Model p$K_a$ |
|---|---|---|---|
| D33 | 5.48 | 5.43 | 3.8 |
| D121 | 4.46 | 4.46 | 3.8 |
| D123 | 4.75 | 4.75 | 3.8 |
| D213 | 3.23 | 3.23 | 3.8 |
| D214 | 4.95 | 4.95 | 3.8 |
| D262 | 6.44 | 6.44 | 3.8 |
| D291 | 4.44 | 4.44 | 3.8 |
| D399 | 7.64 | 7.56 | 3.8 |
| D411 | 2.74 | 2.74 | 3.8 |
| D426 | 6.38 | 6.35 | 3.8 |
| D460 | 8.19 | 8.07 | 3.8 |
| E120 | 3.06 | 3.06 | 4.5 |
| E127 | 4.77 | 4.77 | 4.5 |
| E215 | 4.27 | 4.27 | 4.5 |
| E216 | 3.91 | 3.91 | 4.5 |
| E244 | 4.52 | 4.52 | 4.5 |
| E278 | 4.22 | 4.22 | 4.5 |
| **E312** | 6.71 | 7.46 | 4.5 |
| E321 | 4.53 | 4.53 | 4.5 |
| K138 | 9.86 | 10.04 | 10.5 |
| R189 | 9.10 | 10.02 | 12.5 |
| **TBZ** | 8.57 | | 10 |

The cryo-EM structure (with or without TBZ) was used for p$K_a$ calculations using PROPKA 3.5.0. Key residues of interest are written in boldface.

**Table 5.** Molecular dynamics (MD) simulations of vesicular monoamine transporter 2 (VMAT2) in the presence of dopamine and the observed events.

| System # | Protonated residues | Duration (ns) | Waters in binding pocket* | VMAT2 RMSD[†] from cryo-EM (Å) | Observed events |
|---|---|---|---|---|---|
| DA_0 | None | 2×100 | 7.7±1.5 | 2.6±0.2 | Salt bridge formation between DA and D399; influx of water into the binding pocket. |
| DA_1 | D399 | 2×100 | 10.5±2.2 | 2.2±0.2 | Salt bridges formation between DA and E312; influx of water into the binding pocket. |
| DA_2 | E312, D399 | 2×100 | 8.8±2.1 | 2.2±0.3 | Fluctuations in DA binding pose while remaining within the binding pocket. Formation of two water wires. |
| DA_3 | E312, D399, D426 | 2×100 | 8.8±2.1 | 2.5±0.2 | Fluctuations in DA binding pose; dislocation of DA from its binding site in one of the two simulations. |
| DA_4 | E312, D399, D426, D33 | 1×100 1×200 | 11.7±2.3 | 2.6±0.3 | Fluctuations in DA binding pose; dislocation and release of DA in the 200 ns run. |

*Number of water molecules within 3.5 Å of dopamine averaged over MD snapshots recorded in the time interval 50–100 ns.
[†]RMSDs (root-mean-square-deviations) in VMAT2 Cα-atoms positions; all averages and standard deviations refer to the portion 50–100 ns of MD simulations.

of DA was primarily facilitated by K138, D33 or D426, or N34, rather than R189. This resulted in the exposure of R189, triggering a continuous water pathway near the hydrophobic gate residue F135 (*Figure 5a*). This water path recurred in multiple runs conducted with DA and was lined by the hydrophilic network composed of Y182, R189, Q192, S137, K138, D33, N34, Q142, N146, S200, and D426 (*Figure 5—figure supplement 1d*).

Second, an additional water wire near the hydrophobic gate residue F334 was observed in DA-bound VMAT2 (*Figure 5a*), but not in TBZ-bound VMAT2 which, in contrast, was minimally hydrated (*Figure 5b*). The second water wire was lined by the backbone polar groups of hydrophobic residues, e.g., F334, together with a hydrophilic network containing K379, N388, S338, E312, Y433, Y341, N305, and D399. We also observed that the amine group of DA acted to facilitate the water influx from the lumenal side.

We hypothesize that those two water paths may be related to proton transfer and be associated with protonating the pocket-lining acidic residues. Likely, lumenal DA release depends on the number of protonated acidic residues (*Figure 5—figure supplement 1d–h*). When at least two acidic residues were protonated, we observed fluctuations in DA position (run DA_2; *Video 4*) and dislocation of DA from its pocket (run DA_3; *Video 5*). In system DA_4 (*Table 5*), the protonation of E312, D399, D426, and D33 resulted in complete opening of the hydrophobic gates formed by F135, W318, and F334, which led to the release of DA to the vesicle lumen (*Figure 5c*, *Figure 5—video 1*). DA was observed to migrate toward a cluster of acidic residues, including E127, E120, D121, and D123, before complete dissociation, and the acidic environment within the vesicle lumen should assist in promoting the release of DA.

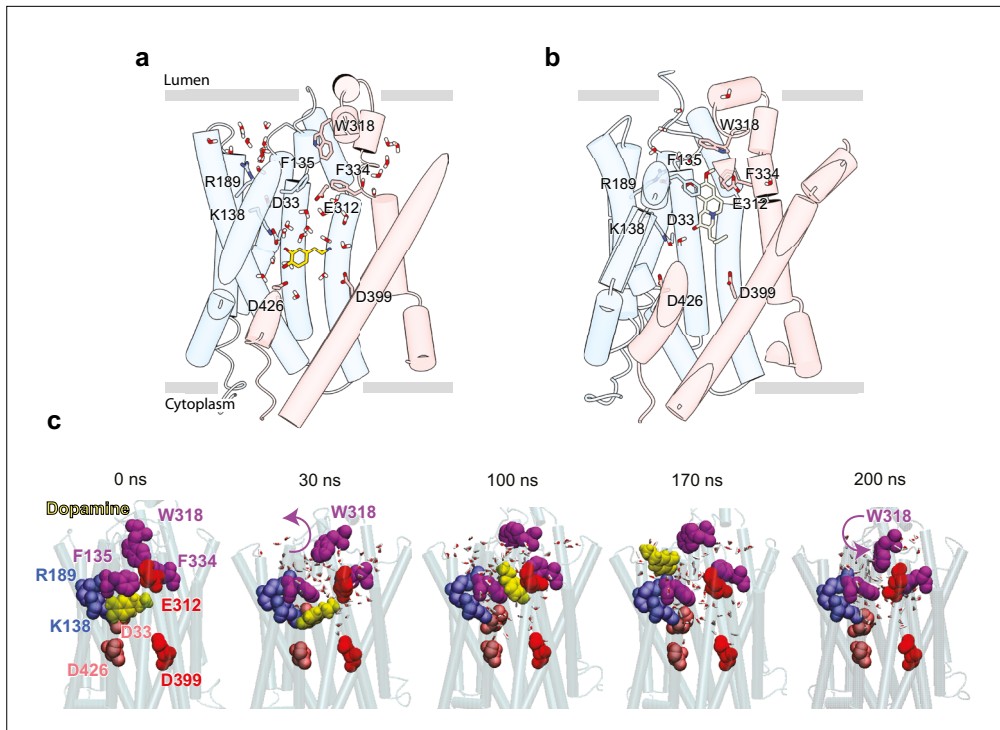

**Figure 5.** Release of dopamine (DA) from the occluded binding site to the lumen through the lumen-facing vestibule and concurrent conformational changes in vesicular monoamine transporter 2 (VMAT2). Comparison of (**a**) DA (*yellow* sticks) and (**b**) tetrabenazine (TBZ) (green sticks) binding to VMAT2, captured by molecular dynamics (MD) simulations. Two water pathways are observed in the MD simulations of VMAT2 bound to DA (run DA_2 in *Table 5*). Water molecules and key residues are shown in sticks. (**c**) Positions of DA (*yellow* van der Waals [VDW] spheres) with respect to hydrophobic gate composed of F135, W318, and F334 (*purple* VDW spheres), and charged residues lining the binding pocket at t=0, 30, 100, 170, and 200 ns. W318 side chain isomerization plays a critical role in mediating the opening/closure of the hydrophobic gate, accompanied by the reorientation of F135 side chain, permitting a flux of water molecules eventually giving rise to the destabilization and release of DA to the synaptic vesicle (SV) lumen by translocating through a hydrated channel. Waters within 10 Å radius from the center of mass (COM) of the hydrophobic gate residues are displayed.

The online version of this article includes the following video and figure supplement(s) for figure 5:

**Figure supplement 1.** Comparisons of binding poses of substrate (dopamine [DA] and serotonin) and inhibitor (tetrabenazine [TBZ]) from docking simulations and molecular dynamics (MD) simulations under different protonation states of substrate-coordinating and/or gating residues.

**Figure 5—video 1.** Release of dopamine (DA) to the vesicular lumen (yellow van der Waals [VDW] spheres), observed after protonating D33, E312, D399, and D426 (red vdW spheres). Related to *Figure 5*.
https://elifesciences.org/articles/91973/figures#fig5video1

## Discussion

The VMAT2-TBZ complex captures the transporter in a fully occluded state with the ligand binding site centrally located between the two repeated substructures TM1–6 and TM7–12. VMAT2 functions by alternating access which involves alternate exposure of the primary binding site to either side of the membrane and isomerization between a cytosolic-open and lumenal-open state in a mechanism known as the rocker-switch (*Figure 6a and b*; *Eiden and Weihe, 2011*; *Radestock and Forrest, 2011*; *Drew et al., 2021*). Studies have proposed that TBZ first enters VMAT2 from the lumenal side, binding to a lumenal-open conformation (*Scherman et al., 1983*). TBZ makes extensive contact with residues in the primary site, likely in a lower affinity state as the transporter subsequently closes to forming the high-affinity occluded state. The lumenal gates lock the transporter into an occluded state, preventing displacement by other ligands and producing a so-called dead-end complex (*Yaffe et al., 2016*; *Kaur et al., 2016*; *Scherman et al., 1983*; *Chen et al., 2012*; *Figure 6a*).

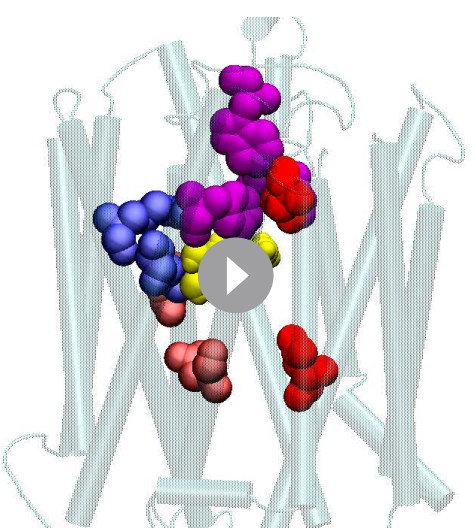

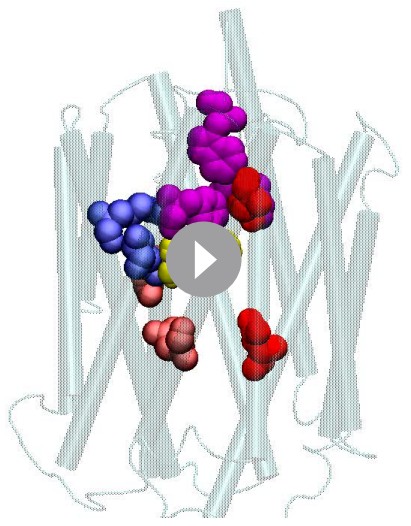

**Video 4.** Fluctuations in dopamine binding pose (DA; *yellow* van der Waals [vdW] spheres) within its binding pocket, observed in a 100 ns molecular dynamics (MD) simulation of vesicular monoamine transporter 2 (VMAT2) in the presence of DA (system DA_2 in *Table 5*). Related to *Table 5*. VMAT2 E312 and D399 are protonated; and D33 and D426 are deprotonated. Hydrophobic gate residues F135, W318, and F334 are displayed in *purple* vdW spheres, and acidic and basic residues K138 and R189 are shown in *red* and *blue* vdW representation.

https://elifesciences.org/articles/91973/figures#video4

**Video 5.** Release of dopamine (DA; yellow van der Waals [vdW] spheres) from the binding pocket, observed in a 100 ns molecular dynamics (MD) simulation (system DA_3 in *Table 4*). (Related to *Table 5*). E312, D399, and D426 were protonated and D33 was deprotonated.

https://elifesciences.org/articles/91973/figures#video5

Comparison of the TMD with more distantly related MFS transporters in other conformational states such as the outward-open VGLUT2 (*Li et al., 2020*) and inward-open GLUT4 (*Yuan et al., 2022*) models (1.2 Å RMSD) shows that the conformational changes involving TM1, -7, -8, and -11 are likely involved in mediating the transport cycle and alternating access (*Figure 6—figure supplement 1*). Comparison with the AlphaFold model shows that while the TMD is largely similar (1.1 Å RMSD overall difference in TMD), AlphaFold lacks several key features such as the conformations of the LLs and is unable to predict key details that are critical to ligand binding. Hence, computational docking could not identify the TBZ binding site using the AlphaFold-predicted model, alluding to the critical importance of our experimental structural data (and simulations based on that structure) for gaining insights into VMAT2 functional mechanisms.

To enable us to solve the structure of VMAT2, we developed an intrinsic fiducial tool consisting of mVenus and the anti-GFP Nb which provides a feature on the cytosolic side of the transporter. GFP and the GFP-Nb are ubiquitous tools used for cell biology, protein biochemistry, and structural biology and here we describe another application of this powerful toolkit. In contrast to traditional antibody-based campaigns which may require multiple rounds of screening and take many months or years to discover a suitable binder, our strategy only required us to make between 15 and 20 constructs to find a suitable chimera (*Figure 1—figure supplement 1a–c*). There are now many other similar methods such as the fusion of the target protein with BRIL and binding of the anti-BRIL Fab (*Mukherjee et al., 2020*) and fusions of maltose-binding protein and DARPin (*Chen et al., 2023*). However, since our strategy directly incorporates fluorescent proteins, this enables FSEC-based screening of constructs and monitoring fluorescence throughout purification (*Hattori et al., 2012*). One obvious disadvantage of all intrinsic fiducial strategies is that the fusion may distort the protein structure, but monitoring the functional activity and comparison with the wild-type protein should mitigate these problems.

Among human mutations in VMAT2 causing an infantile-onset form of parkinsonism (*Jacobsen et al., 2016*; *Rilstone et al., 2013*; *Padmakumar et al., 2019*), three mutants, P316A, P237H, and P387L, have been shown to extinguish monoamine transport. We find these residues are in LL4 and the lumenal ends of TM5 and -9, respectively (*Figure 6—figure supplement 2*). LL4 is involved in lumenal gating and the P316A variant would likely disrupt the conformation of the loop. In the case of P237H, a histidine would result in not only an insertion of a positively charged residue into the lumenal

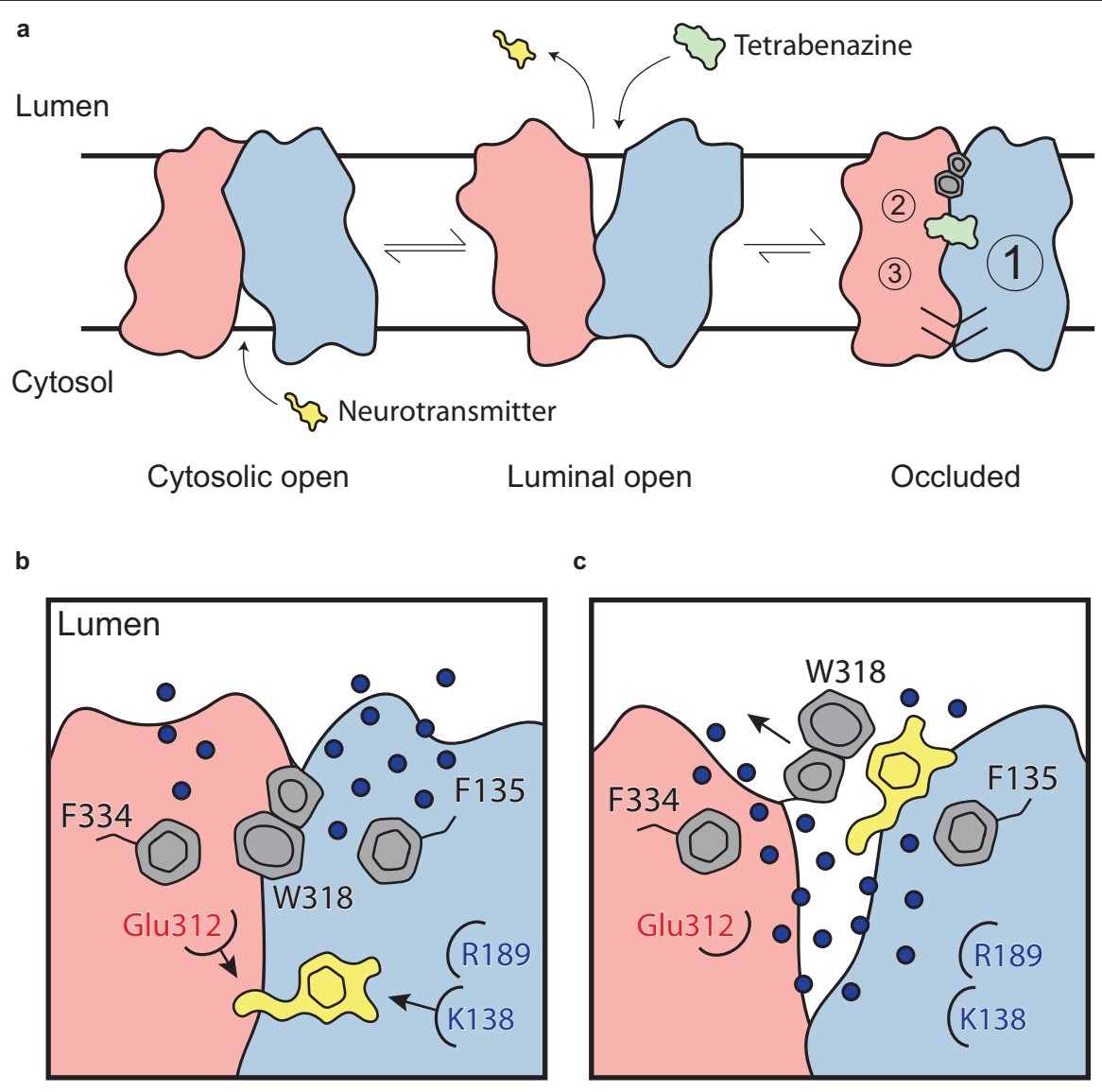

**Figure 6.** Mechanism of tetrabenazine inhibition, and the roles of gating residues and polar interactions networks. (**a**) Cartoon depicting substrate transport and tetrabenazine binding to vesicular monoamine transporter 2 (VMAT2). Neurotransmitter (yellow cartoon) binds to the cytosolic-open conformation before being released from the transporter in the lumenal-open state. Tetrabenazine (green cartoon) binds to the lumenal-facing state and induces a conformational change to a high-affinity occluded conformation which is the resolved cryo-electron microscopy (cryo-EM) structure reported in this work. The VMAT2–tetrabenazine complex highlighting significant features including both cytosolic (slashes) and lumenal gates (hexagon and pentagon depicting W318), the three polar networks (numbered circles) and relative location of the tetrabenazine binding site (green). (**b**) Water penetrating through two pathways is involved in opening the lumenal gate. Dopamine is shown in yellow cartoon. (**c**) Opening of W318 is associated with neurotransmitter release.

The online version of this article includes the following figure supplement(s) for figure 6:

**Figure supplement 1.** Comparison of the vesicular monoamine transporter 2 (VMAT2)-tetrabenazine (TBZ) structure with the predicted AlphaFold structure and with other major facilitator superfamily (MFS) transporters.

**Figure supplement 2.** Human variants of vesicular monoamine transporter 2 (VMAT2).

membrane interface but also in a reduction of the helical bend and would distort the connection of TM5 with TM6. The P387L variant would also disrupt helical connections and the overall architecture of the helix by insertion of a bulky residue into a small hydrophobic cavity. Therefore, we speculate that these SNPs alter the transporter's ability to sample multiple conformations by suppressing transporter dynamics and perturbing VMAT2 stability and folding. Recently, many additional disease

variants have been discovered (*Saida et al., 2023*), many of which are also found in the lumenal or cytoplasmic ends of TM helices, LL1, and the N- and C-termini, and our structure will provide insight into the functional and structural consequences of these variants.

Large aromatic side chains of the lumenal gating residues compartmentalize the transporter, likely ensuring directional transport of substrate. MD simulations revealed little variation in the pose of the aromatic gating residues comprising the inner gates (*Figure 4—figure supplement 2d*). It is interesting to note that substitution of Y433 with alanine resulted in enhanced transport while reducing DTBZ binding. Reduced inhibitor binding is likely due to disruption of π-stacking interactions, but the mechanism for enhanced transport remains unclear. The lumenal gates showed more movement in MD simulations of DA, likely owing to its residue composition and lack of strong interactions (*Figure 5*). Despite this observation, we assert that the tight hydrophobic environment prevents exchange into the lumenal space. These mechanisms of gating are atypical of MFS transporters which more commonly use salt bridges to gate access to the substrate binding site (*Drew et al., 2021*).

Of particular interest among gating residues is W318, which acts as a central plug to prevent solvent access to the transporter. Our mutagenesis analysis suggests that while a large residue is required for maintaining some transporter function, lack of charge also appears to be essential for inhibitor binding. Replacement of W318 with residues present in VMAT2 from other species, except for the phenylalanine mutant, failed to recapitulate any DTBZ binding. Interestingly, of these substitutions, only an arginine was able to fully recapitulate native transport, suggesting lumenal gating activity requires either a large, charged or a large aromatic residue. Our studies also suggest that E127 plays a role in lumenal gating by interacting with LL4 as substitution to alanine-reduced transport, and we speculate that large, charged groups at position 318 directly interact with E127 to maintain full transporter function. In contrast, DBTZ binding remained unaffected for the E127A mutant, suggesting a role for this residue in a later step of transporter closure of the lumenal compartment. Positively charged residues at position 318 may stabilize the lumenal gate and promote conformational cycling through enhanced interactions with negatively charged residues such as E127 but may prevent TBZ from entering the lumenal pathway of the transporter in any capacity. Conservation of this residue in the SLC18 family suggests that this is a common mechanism for lumenal gating. It is unclear what other structural adaptations are present to enable W318 replacement with large positively charged residues in other VMAT2 sequences, but we suspect that the overall architecture of the lumenal domain remains similar. We also note that the disulfide bond between LL1 and -4 plays a critical role in transport (*Yao and Hersh, 2007*), and the disulfide bond may function to restrict the dynamics of this region to allow W318 to occlude the neurotransmitter binding site during transport.

Residues R189 and K138 stand out as critical for drug binding and inhibition and substrate transport. Simulations highlight the role K138 may play in coordinating with the hydroxy group of neurotransmitters and enabling binding at the central site. These results are consistent with previously observed results demonstrating the critical role K138 plays in substrate transport (*Yaffe et al., 2013*; *Merickel et al., 1997*). Remarkably, even mutation to lysine at position 189 abolishes DTBZ binding while still retaining a small amount of transport activity, highlighting the specificity of the DTBZ binding site.

Our structure also provides important clues for understanding the chemical specificity and selectivity of TBZ binding, suggesting that the enhanced affinity of DTBZ is due to preferential interaction with N34. DTBZ is a metabolite of TBZ (*Chen et al., 2012*) which differs by only a hydroxyl group vs. a double bonded oxygen and binds to VMAT2 with approximately twofold higher affinity (*Saida et al., 2023*; *Yao et al., 2011*; *Figure 4a*). Our structure suggests that the amide of N34 acts as a hydrogen bond donor for TBZ, and in the case of DTBZ the hydroxyl of the ligand may act as a hydrogen bond donor for the carbonyl oxygen of N34. We hypothesize that this interaction is more favorable for DTBZ, leading to a higher binding affinity. Valbenazine, a TBZ analogue with a valine attached to the oxygen of the hydroxyl group of DTBZ, binds VMAT2 with a $K_d$ of 150 nM (*Ugolev et al., 2013*; *Uhlyar and Rey, 2018*). We hypothesize that N34 does not form a favorable hydrogen bond with the oxygen of valbenazine and that addition of this larger moiety causes steric clashes in the binding site. Mutation of N34 nearly eliminated DTBZ binding, highlighting its importance in coordinating inhibitor binding. We found N34 is also important for transport, but its precise role remains unclear. Strikingly, replacement with glutamine had no effect on transport whereas aspartate had only a small effect. This suggests that N34 is likely involved in substrate coordination but may not be entirely essential. This is confirmed with the N34A substitution which still showed significant transport activity. In contrast,

a threonine substitution was greatly detrimental, but we believe this is likely due to larger active site perturbations.

Comparison of the residues involved in TBZ binding in VMAT2 vs. VMAT1 also provides insight into the selectivity of TBZ by demonstrating that key differences in the ligand binding site are likely responsible for the reduction in TBZ binding affinity observed in VMAT1 (*Peter et al., 1994*). L37 and V232 are highlighted, and mutation to the equivalent residue in VMAT1, phenylalanine, and leucine, respectively, eliminated DTBZ binding but only reduced substrate transport. Given these results, we suggest these are the primary residues involved in determining TBZ selectivity for VMAT2 vs. VMAT1. We suspect VMAT1 exhibits structural differences compared with VMAT2 to accommodate the larger residues and maintain functional activity.

Through MD simulations, we gained insights into the effects of the protonation states of TBZ and E312. Protonation of TBZ at the amine resulted in an unstable complex, and a structure markedly different from the state resolved with electron microscopy. This suggests TBZ remains unprotonated when bound to the occluded state, as the protonated form will not remain in complex with VMAT2. We believe there may be other TBZ-bound states such as the lumenal-open state where TBZ may be protonated. However, in the terminal complex, E312 must remain protonated suggesting that E312 acts as a hydrogen bond donor to stabilize TBZ in the binding site. Our binding and transport studies further confirm this as the mutation of E312 to glutamine greatly reduces serotonin transport and DTBZ binding (*Figure 4f and g*). Moreover, mutation of E312 to an alanine has long been known to completely negate all TBZ binding and transport activities (*Yaffe et al., 2013*; *Støve et al., 2022*) again highlighting the requirement of hydrogen bond pairing with TBZ.

We also highlight three key polar networks which may be involved in conformational changes induced by proton binding during the transport cycle and are likely also involved in mediating proton transduction (*Figure 6a*). Taken together, we believe these networks play a critical role in mediating the conformation changes taking place in the transporter upon substrate or inhibitor binding. We hypothesize that the protonation of D33, E312, D399, and D426 would significantly perturb these interactions by breaking crucial hydrogen bond pairs and/or salt bridges, leading to opening of the cytosolic gate. This falls in line with previous work highlighting the role these residues play in transport through a variety of mutagenesis and functional experiments (*Merickel et al., 1997*; *Støve et al., 2022*). Given the known transport stoichiometry of two protons per neurotransmitter, we speculate that two protons may dissociate back into the lumen, perhaps driven by the formation of salt bridges between D33 and K138 or R189 and E312 for example in an cytosol-facing state. The asymmetry between these two networks is also striking, with the first network consisting of TM1, -4, and -11 being substantially larger. This may allow for larger conformational changes on this side and an overall asymmetry in the cytosolic-open state of the transporter. To our knowledge, this is also an atypical feature in MFS proteins (*Drew et al., 2021*) and would represent an interesting adaptation upon the rocker-switch mechanism.

Our studies lead us to propose a mechanism by which TBZ inhibits VMAT2 by first entering the transporter from the lumenal-open state and subsequently trapping it in an occluded state through a two-step mechanism centered around W318, which acts as a central plug to close off the lumenal compartment. This provides a basis for non-competitive inhibition as endogenous neurotransmitter is unable to compete for binding from the closed off cytosolic compartment (*Figure 6a*). Our results also provide insight into transporter function, and we propose a mechanism for substrate release from the central binding site (*Figure 6b*). The bound substrate coordinates in the binding site predominantly through interactions with E312 and K138. R189 is directed toward the lumenal space where it engages with one of two water channels. Water enters through these channels into the transporter, passing F135 and F334 and inducing conformational change. In our simulations, we find intrusion of water to cause W318 to flip out of the central site which enables neurotransmitter to dissociate from the transporter (*Figure 6b*).

In summary, we have developed a new fiducial tool incorporating mVenus and the GFP-Nb into the primary sequence of a previously intractable transport protein of great importance to human health. Cryo-EM of VMAT2 allowed us to pinpoint the lumenal and cytoplasmic gates, polar networks likely involved in conformational change and proton transduction, and the conformation and binding site of inhibitors. Our structure facilitated the discovery and detailed analysis of many residues involved in these key molecular mechanisms and enabled further extension in our understanding of

neurotransmitter transport. Thus, by capturing the VMAT2 bound with TBZ along with subsequent mutagenesis, binding, transport, and computational experiments, our work delivers insights into the transporter's mechanism of function and provides a framework for understanding the structural underpinnings of neurotransmitter release, transport, and inhibition in VMAT2 and other related transport proteins in the MFS family.

# Methods

**Key resources table**

| Reagent type (species) or resource | Designation | Source or reference | Identifiers | Additional information |
|---|---|---|---|---|
| Gene (*Homo sapiens*) | Human vesicular monoamine transporter 2 | Clone ID 40025175 | NCBI Reference Sequence: BC108928 | Horizon Discovery |
| Gene (*Lama glama*) | Anti-GFP nanobody | Plasmid #49172 | | Addgene |
| Cell line (*Homo sapiens*) | HEK293S GnTI⁻ | ATCC | Cat # ATCC CRL-3022 | Used for expression of VMAT2 (PMID:27929454) |
| Cell line (*Spodoptera frugiperda*) | Sf9 | ATCC | Cat # ATCC CRL-1711 | Used in production of baculovirus for transduction (PMID:27929454) |
| Transfected construct (human) | pEG BacMam | Gouaux lab | | PMID:25299155 |
| Affinity chromatography resin | Strep-Tactin Superflow high capacity resin | IBA Lifesciences | Cat # 2-1208-500 | Affinity purification resin |
| Chemical compound, drug | *n*-Dodecyl-β-D-maltoside | Anatrace | Cat # D310 | Detergent |
| Chemical compound, drug | Cholesteryl hemisuccinate | Anatrace | Cat # CH210 | Lipid |
| Chemical compound, drug | Reserpine | Sigma | Cat # 83580 | Inhibitor |
| Chemical compound, drug | Dihydrotetrabenazine | Cayman Chemicals | Cat # 27182 | Inhibitor |
| Chemical compound, drug | [³H]5-HT | PerkinElmer | Cat # NET1167250UC | Radiolabeled substrate |
| Chemical compound, drug | [³H]dihydrotetrabenazine | American Radiolabeled Chemicals | Cat # ART 2119 | Radiolabeled inhibitor |
| Chemical compound, drug | Tetrabenazine | Sigma | Cat # T2952 | Inhibitor |
| Chemical compound, detergent | Digitonin | Sigma | Cat # 300410 | Detergent |
| Software, algorithm | Phenix | PMID:22505256 | RRID:SCR_014224 | https://www.phenix-online.org/ |
| Software, algorithm | SerialEM | PMID:16182563 | RRID:SCR_017293 | http://bio3d.colorado.edu/SerialEM |
| Software, algorithm | CryoSPARC | PMID:28165473 | RRID:SCR_016501 | https://cryosparc.com/ |
| Software, algorithm | RELION | PMID:23000701 | RRID:SCR_016274 | http://www2.mrc-lmb.cam.ac.uk/relion |
| Software, algorithm | UCSF-Chimera | PMID:15264254 | RRID:SCR_004097 | https://www.cgl.ucsf.edu/chimera/ |
| Software, algorithm | Coot | PMID:15572765 | RRID:SCR_014222 | https://www2.mrc-lmb.cam.ac.uk/personal/pemsley/coot |
| Software, algorithm | MolProbity | PMID:20057044 | RRID:SCR_014226 | http://molprobity.biochem.duke.edu/ |

*Continued on next page*

Continued

| Reagent type (species) or resource | Designation | Source or reference | Identifiers | Additional information |
|---|---|---|---|---|
| Other | R 2/1 200 mesh Au holey carbon grids | Electron Microscopy Sciences | Cat # Q210AR1 | Cryo-EM grids |
| Other | R 1.2/1.3 200 mesh Cu holey carbon grids | Electron Microscopy Sciences | Cat # Q2100CR1.3 | Cryo-EM grids |
| Other | Copper HIS-Tag YSI | PerkinElmer | Cat # RPNQ0096 | SPA beads |

## Data reporting

No statistical methods were used to predetermine sample size. The experiments were not randomized, and the investigators were not blinded to allocation during experiments and outcome assessment.

## VMAT2 construct design and cloning

Wild-type VMAT2 was expressed as a C-terminal eGFP fusion protein containing an 8x His-tag. The VMAT2 chimera consisted of mVenus (*Kremers et al., 2006*; *Rothbauer et al., 2008*) fused to the N-terminus of VMAT2 at amino acid position 17, and the anti-GFP nanobody (*Kubala et al., 2010*; *Rothbauer et al., 2008*) containing both a 10x His-tag and a TwinStrep tag fused to the C-terminus at position 482 by Infusion cloning. Single-point mutants were made using PCR starting from wild-type VMAT2 C-terminally tagged eGFP construct, and constructs were initially evaluated using FSEC (*Hattori et al., 2012*).

## Expression and purification

VMAT2 was expressed in HEK293S GnTI⁻ cells (*Reeves et al., 2002*) using baculovirus-mediated transduction (*Goehring et al., 2014*). Enriched membranes were first isolated by sonication followed by an initial spin at 10,000×$g$ followed by a 100,000×$g$ spin and subsequent homogenization. Membranes resuspended in 25 mM Tris pH 8.0 and 150 mM NaCl and frozen at –80°C until use. Membranes were thawed and solubilized in 20 mM $n$-dodecyl-β-D-maltoside (DDM) and 2.5 mM cholesteryl hemisuccinate (CHS) with 1 mM DTT and 10 µM TBZ for 1 hr before centrifugation at 100,000×$g$. VMAT2 was purified into buffer containing 1 mM DDM, 0.125 mM CHS, 25 mM Tris, 150 mM NaCl, 1 mM DTT, and 1 µM TBZ pH 8.0 using either a NiNTA column which was eluted in the same buffer containing 250 mM imidazole (for the wild-type VMAT2 C-terminally tagged eGFP protein) or a StrepTactin column eluted with 5 mM desthiobiotin. Purified VMAT2 was pooled and concentrated using a 100 kDa concentrator (Amicon) before separating by size-exclusion chromatography on a Superose 6 Increase column in 1 mM DDM, 0.125 mM CHS, 25 mM Tris pH 8.0, 150 mM NaCl, 1 mM DTT, and 1 µM TBZ. Peak fractions were pooled, concentrated to 6 mg/ml with a 100 kDa concentrator before addition of 100 µM TBZ, and ultracentrifuged at 60,000×$g$ prior to cryo-EM grid preparation.

## Cryo-EM sample preparation and data acquisition

The VMAT2 chimera (1.5 µl) at a concentration of 6 mg/ml was applied to glow discharged Quantifoil holey carbon grids (1.2/1.3 or 2/1 200 mesh copper or gold). Grids were blotted for 4 s at 100% humidity, 4°C, with a blot force of 4 using a Vitrobot Mk IV (Thermo Fisher) before flash freezing into a 50/50 mixture of liquid propane/ethane cooled to ~170°C with liquid nitrogen. Videos containing 40 frames were recorded on a FEI Titan Krios operating at 300 kV equipped with a Gatan K3 direct electron detector and a Bioquantum energy filter set to a slit width of 20 eV. Images were collected in super-resolution counting mode at a pixel size of 0.647 Å/pixel with defocus ranges from –1 to –2.5 µm with a total dose of 60 e⁻/Å². Images were recorded using SerialEM (*Mastronarde, 2003*).

## Image processing

Micrographs were corrected using Patch Motion Correction and contrast transfer function estimated using Patch CTF in CryoSPARC v4.2 (*Punjani et al., 2017*). A total of 24,875 micrographs were collected between two datasets recorded on the same microscope. Particles were initially classified by 2D classification in CryoSPARC to generate an ab initio model for template picking which resulted in ~10 million picks which were extracted at a box size of 384 binned to 128 and classified

multiple times using 2D classification and hetero-refinement using a newly generated ab initio model, an empty detergent 'decoy' class, and a junk class containing random density. The resulting approximately 500,000 particles from each dataset were re-extracted at a box size of 384 binned to 256, refined using non-uniform refinement (*Punjani et al., 2020*), and combined before being subjected to further classification and refinement. The remaining 212k particles were then re-extracted at a full box size of 384, refined, and subjected to Bayesian polishing in RELION 3.1 (*Scherman et al., 1983*; *Scheres, 2012*). The resulting 3.5 Å map still exhibited significant anisotropy and was subjected to further rounds of 3D classification and refinement in CryoSPARC with a TMD mask to improve features of the peripheral TMs. The final 65,516 particles resulted in a 3.12 Å map that was sharpened in Deep-EMhancer (*Sanchez-Garcia et al., 2021*) using the mask setting and utilizing a mask around the TMD only.

## Model building

The resulting EM map was sufficient for modeling most VMAT2 side chains in the TMD. The Alpha-Fold2 (*Jumper et al., 2021*) model of VMAT2 was used for initial fitting and was further refined using RosettaCM (*Wang et al., 2016*). After successive rounds of RosettaCM, the model was locally fit using Coot 0.98 (*Emsley and Cowtan, 2004*) and Isolde (*Croll, 2018*) with the majority of the manual rebuilding being done in Isolde. The model was refined in real space using Phenix 1.2 (*Liebschner et al., 2019*) and validated by comparing the FSC between the half maps and the refined model.

## Radioligand binding assays

Purified VMAT2 (wild-type eGFP tagged and VMAT2 chimera) were diluted to 5 nM in 1 mM DDM, 0.125 mM CHS in 20 mM Tris pH 8.0, 150 mM NaCl with 1 mg/ml CuYSi beads (PerkinElmer). Protein concentration was estimated using FSEC and a standard GFP concentration curve. Protein was then mixed 1:1 to a final protein concentration of 2.5 nM in detergent buffer containing serially diluted $^3$H-labeled DTBZ (American Radiolabeled Chemicals) starting at 60 nM final concentration. Counts were then measured using a Microbeta2 scintillation counter in 96-well plates with triplicate measurements (*Green et al., 2015*). Specific counts were obtained by subtracting values obtained by the addition of 100 µM reserpine. Mutants were evaluated similarly from cell lysates of transfected cells. Data were fit to a single-site binding equation using GraphPad Prism.

Competition binding experiments were performed at the same protein concentration in the same detergent buffer. 10 nM $^3$H-labeled DTBZ was added to buffer and used for nine 1:1 serial dilutions with detergent buffer which initially contained 100 µM reserpine (10 µM for chimera). Measurements were done in triplicates and fit with a one-site competitive binding equation in GraphPad Prism.

## Serotonin transport

Cells transduced overnight were spun down and resuspended in 140 mM KCl, 5 mM MgCl$_2$, 50 mM HEPES-KOH pH 7.4, and 0.3 M sucrose. Cells were permeabilized at 30°C for 10 min in the presence of 5 mM MgATP and 0.01% digitonin (*Yaffe et al., 2016*). Controls additionally included 100 µM reserpine. After 10 min, cells were spun down and resuspended in the same buffer containing 2.5 mM MgATP and incubated at 30°C for 10 min. Cells were then mixed 1:1 with buffer containing $^3$H-labeled serotonin at a final concentration of 1 or 10 µM and incubated at room temperature for 6 min. Transport was stopped by the addition of ice-cold buffer, and the cells were collected on Glass Fiber C filters. The filters were then counted in scintillation fluid. Time course experiments were performed in the same way using 1 µM serotonin.

To evaluate mutants, ~5 million cells were transfected with 5 µg of DNA using Polyjet reagent. 24 hr after transfection, the cells were harvested into assay buffer containing 130 mM KCl, 5 mM MgCl$_2$, 25 mM HEPES-KOH pH 7.4, 1 mM ascorbic acid, 5 mM glucose, and washed once with 1 ml of assay buffer. Cells were permeabilized in 500 µl of assay buffer containing 0.001% digitonin and washed once with 1 ml of assay buffer. Next the cells were resuspended, mixed 1:1 with of 0.2 µM $^3$H-labeled serotonin with 5 mM ATP in assay buffer, and incubated at room temperature for 15 min. In some cases, 12.5 µM reserpine was added to the cells along with $^3$H-labeled serotonin for a control. Cells were washed with 2 ml of ice-cold assay buffer and solubilized with 5% Triton X-100 followed by scintillation counting. Mutant expression was assessed with fluorescent microscopy to ensure protein expression levels were comparable.

## MD simulations

The initial MD simulation system was prepared using CHARMM-GUI Membrane Builder module (*Wu et al., 2014*). The structure of VMAT2 bound to TBZ was aligned using PPM2.0 webserver (*Lomize et al., 2012*) and embedded into 1-palmitoyl-2-oleoyl-*sn*-glycero-3-phosphocholine (POPC) membrane lipids. The p$K_a$ values of titratable residues of VMAT2 in the presence or absence of TBZ were calculated using PROPKA 3.5.0 (*Olsson et al., 2011*); and the computed p$K_a$ values for acidic residues and TBZ are listed in *Table 4*. Of note, D33 and E312, as well as D399 and D426 (yellow highlights in *Figure 5—figure supplement 1c* and bold in *Table 4*) are in proximity to TBZ; thus, their protonation states may impact substantially TBZ binding. Given that D33 may form a salt bridge with its nearby K138, we assumed it to be deprotonated; and E312 and D399 were assumed to be protonated because of their comparatively higher p$K_a$ values. To further assess how the protonation states on TBZ and/or D426 affect the binding propensity of TBZ, we constructed four different simulation systems (see *Table 3*) with either protonated or deprotonated TBZ and with/without protonation of D426 (in addition to E312 and D399), denoted as systems TBZ_1–TBZ_4. For each system, TIP3P waters and $K^+$ and $Cl^-$ ions corresponding to 0.1 M KCl solution were added to build a simulation box of 92×92×108 Å$^3$. The simulated systems contained approximately 86,000 atoms, including VMAT2 and TBZ, 203 POPC molecules, and 17,400 water molecules.

All MD simulations were performed using NAMD (*Phillips et al., 2005*) (version NAMD_2.13), following default protocol and parameters implemented in CHARMM-GUI (*Lee et al., 2016*). Briefly, CHARMM36 force fields were adopted for VMAT2, lipids, and water molecules (*Klauda et al., 2010*; *Huang et al., 2017*). Force field parameters for both protonated (carried +1 charge) and neutral-charged TBZ were obtained from the CHARMM General Force Field for drug-like molecules (*Vanommeslaeghe et al., 2012*); and proton was added using Open Babel (*O'Boyle et al., 2011*). Prior to productive runs, the simulation systems were energy-minimized for 10,000 steps, followed by 2 ns (*Hoover, 1985*; *Nosé, 1984*) constant pressure (1 bar, 310 K; NPT) simulation during which the constraints on the protein backbone were reduced from 10 to 0 kcal/mol. Finally, the unconstrained protein was subjected to 100–180 ns NPT simulations. Periodic boundary conditions were employed in all simulations, and the particle mesh Ewald method (*Essmann et al., 1995*) was used for long-range electrostatic interactions with the pair list distance of 16.0 Å. The simulation time step was set to 2 fs with the covalent hydrogen bonds constrained with the SHAKE algorithm (*Ryckaert et al., 1977*). A force-based switching function was used for Lennard-Jones interactions with switching distance set to 10 Å. Langevin dynamics was applied with a piston period of 50 fs and a piston decay of 25 fs, as well as Langevin temperature coupling with a friction coefficient of 1 ps$^{-1}$. For each system, two to three independent runs of 100–180 ns were performed (see *Table 3*). Snapshots from trajectories were recorded every 100 ps for statistical analysis interactions and structural changes.

For DA bound to the occluded VMAT2, we constructed five simulation systems with varying protonation states for the four pocket-lining acidic residues (denoted as runs DA_0 to DA_4; see *Table 5*). In all systems, the DA carried +1 charge and was initially positioned at the top (lowest energy) docking-predicted binding pose (see *Figure 5—figure supplement 1*); and for each system, two runs of 100 ns were performed, following the same simulation protocols described above for TBZ, except for an extended run of 200 ns conducted to visualize the DA release to the SV lumen, which comprised 170 ns conventional MD simulations, followed by 4 ns of enhanced conformation sampling using the adaptive biasing force method (*Cheng et al., 2018*; *Chipot and Hénin, 2005*), and subsequent 30 ns conventional MD simulations.

## Docking simulations

The binding of DA, serotonin, and TBZ to the AlphaFold2-modeled VMAT2 conformer (AF-Q05940-F1-model_v4) and to the present cryo-EM-resolved structure were simulated using AutoDock Vina (*Trott and Olson, 2010*). The molecular structures of protonated DA and serotonin were adopted from the previous studies (*Cheng et al., 2015*; *Yang and Gouaux, 2021*). Docking simulations were carried out using a grid with dimensions set to 65×65×70 Å$^3$ to encapsulate the entire structure of the transporter. The exhaustiveness of the simulation was set to 50, and the algorithm returned a maximum of 20 ligand binding poses.

## Computational data analysis

MD trajectory analysis and visualization were performed using VMD (*Humphrey et al., 1996*). For each snapshot, the TBZ binding affinity was evaluated using contact-based binding affinity predictor PRODIGY-LIG (*Vangone et al., 2019*); and the number of waters within the TBZ binding pocket were assessed by counting the number of oxygen ($OH_2$) atoms within 3.5 Å of TBZ. For RMSD calculations, the $C^\alpha$-atoms of VMAT2 and the heavy atoms from TBZ were used after alignment of the simulated VAMT2 onto the cryo-EM structure. The binding pockets of VMAT2 were assessed using Essential Site Scanning Analysis (ESSA) (*Kaynak et al., 2020*) and Fpocket (*Le Guilloux et al., 2009*), implemented in ProDy 2.0 (*Zhang et al., 2021*). Sequence conservation of VMAT2 was computed using the ConSurf server with default parameters (*Yariv et al., 2023*). Multiple sequence alignment for SLC18 family members was performed using PROMALS3D (*Pei et al., 2008*).

## Acknowledgements

This work was funded by the National Institutes of Health (1R01NS134558 to JAC, and 1R01GM139297 to IB). A portion of this research was supported by NIH grant U24GM129547 and performed at the PNCC at OHSU and accessed through EMSL (grid.436923.9), a DOE Office of Science User Facility sponsored by the Office of Biological and Environmental Research. Microscopy at the University of Pittsburgh was supported by National Institutes of Health grants S10 OD025009 and S10 OD019995.

---

## Additional information

### Funding

| Funder | Grant reference number | Author |
| --- | --- | --- |
| National Institute of Neurological Disorders and Stroke | 1R01NS134558 | Jonathan A Coleman |
| National Institute of General Medical Sciences | 1R01GM139297 | Ivet Bahar |

The funders had no role in study design, data collection and interpretation, or the decision to submit the work for publication.

### Author contributions

Michael P Dalton, Conceptualization, Data curation, Validation, Investigation, Visualization, Methodology, Writing – original draft, Writing – review and editing; Mary Hongying Cheng, Data curation, Validation, Investigation, Visualization, Methodology, Writing – original draft, Writing – review and editing; Ivet Bahar, Supervision, Funding acquisition, Methodology, Project administration, Writing – review and editing; Jonathan A Coleman, Conceptualization, Resources, Data curation, Supervision, Funding acquisition, Validation, Investigation, Visualization, Methodology, Writing – original draft, Project administration, Writing – review and editing

### Author ORCIDs

Michael P Dalton ![ORCID] http://orcid.org/0000-0001-5296-5099
Mary Hongying Cheng ![ORCID] http://orcid.org/0000-0001-5833-8221
Jonathan A Coleman ![ORCID] https://orcid.org/0000-0003-0001-6195

Reviewer #1 (Public Review): https://doi.org/10.7554/eLife.91973.4.sa1
Reviewer #2 (Public Review): https://doi.org/10.7554/eLife.91973.4.sa2
Reviewer #3 (Public Review): https://doi.org/10.7554/eLife.91973.4.sa3
Author Response https://doi.org/10.7554/eLife.91973.4.sa4

## Data availability

The coordinates and associated volumes for the cryo-EM reconstruction of the VMAT2-TBZ data set have been deposited in the Protein Data Bank (PDB) and Electron Microscopy Data Bank (EMDB) under the accession codes 8THR and 41269.

The following datasets were generated:

| Author(s) | Year | Dataset title | Dataset URL | Database and Identifier |
|---|---|---|---|---|
| Dalton MP, Coleman JA | 2023 | Structure of the human vesicular monoamine transporter 2 (VMAT2) bound to tetrabenazine in an occluded conformation | https://www.rcsb.org/structure/8THR | RCSB Protein Data Bank, 8THR |
| Dalton MP, Coleman JA | 2023 | Structure of the human vesicular monoamine transporter 2 (VMAT2) bound to Tetrabenazine in an occluded conformation | https://www.ebi.ac.uk/emdb/EMD-41269 | Electron Microscopy Data Bank, EMD-41269 |

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
