## [Editor Report · eLife assessment]

The report presents the cryo-EM structure of human vesicular monoamine transporter 2 (VMAT2) bound to tetrabenazine, a clinical drug. VMAT2 is critical for neurotransmission, and the study constitutes an **important** milestone in neurotransmitter transport research. The evidence presented in the report is **convincing** and provides new opportunities for developing improved therapeutic interventions and furthering our understanding of this vital protein's function.

---

## [Referee Report · Reviewer #1 (Public Review)]

Summary:

This study presents fundamental new insights into vesicular monoamine transport and the binding pose of the clinical drug tetrabenazine (TBZ) to the mammalian VMAT2 transporter. Specifically, this study reports the first structure for the mammalian VMAT (SLC18) family of vesicular monoamine transporters. It provides insights into the mechanism by which this inhibitor traps VMAT2 into a 'dead-end' conformation. The structure also provides some evidence for a novel gating mechanism within VMAT2, which may have wider implications for understanding the mechanism of transport in the wider SLC18 family.

Strengths:

The structure is high quality, and the method used to determine the structure via fusing mVenus and the anti-GFP nanobody to the amino and carboxyl termini is novel. The binding and transport data are convincing and provide new insights into the role of conserved side chains within the SLC18 members. The binding position of TBZ is of high value, given its role in treating Huntington's chorea and for being a 'dead-end' inhibitor for VMAT2.

---

## [Referee Report · Reviewer #2 (Public Review)]

As a report of the first structure of VMAT2, indeed the first structure of any vesicular monoamine transporter, this manuscript represents an important milestone in the field of neurotransmitter transport. VMAT2 belongs to a large family (the major facilitator superfamily, MFS) containing transporters from all living species. There is a wealth of information relating to the way that MFS transporters bind substrates, undergo conformational changes to transport them across the membrane and couple these events to the transmembrane movement of ions. VMAT2 couples the movement of protons out of synaptic vesicles to the vesicular uptake of biogenic amines (serotonin, dopamine and norepinephrine) from the cytoplasm. The new structure presented in this manuscript can be expected to contribute to an understanding of this proton/amine antiport process.

The structure contains a molecule of the inhibitor TBZ bound in a central cavity, with no access to either luminal or cytoplasmic compartments. The authors carefully analyze which residues interact with bound TBZ and measure TBZ binding to VMAT2 mutated at some of those residues. These measurements allow well-reasoned conclusions about the differences in inhibitor selectivity between VMAT1 and VMAT2 and differences in affinity between TBZ derivatives.

The structure also reveals polar networks within the protein and hydrophobic residues in positions that may allow them to open and close pathways between the central binding site and the cytoplasm or the vesicle lumen. The authors propose involvement of these networks and hydrophobic residues in coupling of transport to proton translocation and conformational changes.

---

## [Referee Report · Reviewer #3 (Public Review)]

Summary:

The vesicular monoamine transporter is a key component in neuronal signaling and is implicated in diseases such as Parkinson's. Understanding of monoamine processing and our ability to target that process therapeutically has been to date provided by structural modeling and extensive biochemical studies. However, structural data is required to establish these findings more firmly.

Strengths:

Dalton et al resolved a structure of VMAT2 in the presence of an important inhibitor, tetrabenazine, with the protein in detergent micelles, using cryo-EM and with the aid of protein domains fused to its N- and C-terminal ends, including one fluorescent protein that facilitated protein screening and purification. The resolution of the maps allows clear assignment of the amino acids in the core of the protein. The structure is in good agreement with a wealth of experimental and structural prediction data, and provides important insights into the binding site for tetrabenazine and selectivity relative to analogous compounds. The authors provide additional biochemical analyses that further support their findings. The comparison with AlphaFold models is enlightening.

---

## [Author Response]

The following is the authors’ response to the previous reviews.

**Public Reviews:**

**Reviewer #1 (Public Review):**
Summary:This study presents fundamental new insights into vesicular monoamine transport and the binding pose of the clinical drug tetrabenazine (TBZ) to the mammalian VMAT2 transporter. Specifically, this study reports the first structure for the mammalian VMAT (SLC18) family of vesicular monoamine transporters. It provides insights into the mechanism by which this inhibitor traps VMAT2 into a 'dead-end' conformation. The structure also provides some evidence for a novel gating mechanism within VMAT2, which may have wider implications for understanding the mechanism of transport in the wider SLC18 family.Strengths:The structure is high quality, and the method used to determine the structure via fusing mVenus and the anti-GFP nanobody to the amino and carboxyl termini is novel. The binding and transport data are convincing and provide new insights into the role of conserved side chains within the SLC18 members. The binding position of TBZ is of high value, given its role in treating Huntington's chorea and for being a 'dead-end' inhibitor for VMAT2.

We thank reviewer #1 for their constructive comments and input which we feel has greatly improved the manuscript.

**Reviewer #2 (Public Review):**

This public review is the same review that was posted earlier and has not been updated in response to our comments or to the revised manuscript. Please see our earlier response to these comments. We thank reviewer #2 for their input and we have incorporated many of these suggestions into our revised manuscript. With regard to the question of ‘how TBZ got there’, we have revised this sentence in the discussion to be more speculative. As pointed out earlier, our interpretation of the structure is based on a wealth of experimental and structural data which support our interpretations. Thus, our conclusions have not been overstated. This has been explained in our earlier public response and these key studies have been cited throughout the manuscript. We also note that reviewer #3 found the AlphaFold comparisons to be quite meaningful.

Overview:As a report of the first structure of VMAT2, indeed the first structure of any vesicular monoamine transporter, this manuscript represents an important milestone in the field of neurotransmitter transport. VMAT2 belongs to a large family (the major facilitator superfamily, MFS) containing transporters from all living species. There is a wealth of information relating to the way that MFS transporters bind substrates, undergo conformational changes to transport them across the membrane and couple these events to the transmembrane movement of ions. VMAT2 couples the movement of protons out of synaptic vesicles to the vesicular uptake of biogenic amines (serotonin, dopamine and norepinephrine) from the cytoplasm. The new structure presented in this manuscript can be expected to contribute to an understanding of this proton/amine antiport process.The structure contains a molecule of the inhibitor TBZ bound in a central cavity, with no access to either luminal or cytoplasmic compartments. The authors carefully analyze which residues interact with bound TBZ and measure TBZ binding to VMAT2 mutated at some of those residues. These measurements allow well-reasoned conclusions about the differences in inhibitor selectivity between VMAT1 and VMAT2 and differences in affinity between TBZ derivatives.The structure also reveals polar networks within the protein and hydrophobic residues in positions that may allow them to open and close pathways between the central binding site and the cytoplasm or the vesicle lumen. The authors propose involvement of these networks and hydrophobic residues in coupling of transport to proton translocation and conformational changes. However, these proposals are quite speculative in the absence of supporting structures and experimentation that would test specific mechanistic details.Critique:Although the structure presented in this MS is clearly important, I feel that the authors have overstated several of the conclusions that can be drawn from it. I don't agree that the structure clearly indicates why TBZ is a non-competitive inhibitor; the proposal that specific hydrophobic residues function as gates will depend on lumen- and cytoplasm-facing structures for verification; the polar networks could have any number of functions - indeed it would be surprising if they were all involved in proton transport. Several of these issues could be resolved by a clearer illustration of the data, but I believe that a more rigorous description of the conclusions and where they fall between firm findings and speculation would help the reader put the results in perspective.Non-competitive inhibition occurs when the action of an inhibitor can't be overcome by increasing substrate concentration. The structure shows TBZ sequestered in the central cavity with no access to either cytoplasm or lumen. The explanation of competitive vs non-competitive inhibition depends entirely on how TBZ got there. If it bound from the cytoplasm, cytoplasmic substrate should have been able to compete with TBZ and overcome the inhibition. If it bound from the lumen, or from within the bilayer, cytoplasmic substrate would not be able to compete, and inhibition would be non-competitive. The structure does not tell us how TBZ got there, only that it was eventually occluded from both aqueous compartments and the bilayer.The issue of how VMAT2 opens access to the central binding site from luminal and cytoplasmic sides is an important and interesting one, and comparison with other MFS structures in cytoplasmic-open or extracellular/luminal-open is a very reasonable approach. However, any conclusions for VMAT2 should be clearly indicated as speculative in the absence of comparable open structures of VMAT2. As a matter of presentation, I found the illustrations in ED Fig. 6 to be less helpful than they could have been. Specifically, illustrations that focus on the proposed gates, comparing that region of the new structure with the corresponding region of either VGLUT or GLUT4 would better help the reader to compare the position of the proposed gate residues with the corresponding region of the open structure. I realize that is the intended purpose of ED Fig. 6b and 6c, but currently, those show the entire protein and a focus on the gate regions might make the proposed gate movements clearer. I also appreciate the difference between the Alphafold prediction and the new structure, but I'm not convinced that ED Fig. 6a adds anything helpful.The polar networks described in the manuscript provide interesting possibilities for interactions with substrates and protons whose binding to VMAT2 must control conformational change. Aside from the description of these networks, there is little evidence presented to assess the role of these networks in transport. Are the networks conserved in other closely related transporters? How could the interaction of the networks with substrate or protons affect conformational change? Of course, any potential role proposed for the networks would be highly speculative at this point, and any discussion of their role should point out their speculative nature and the need for experimental verification. Some speculation, however, can be useful for focusing the field's attention on future directions. However, statements in the abstract (three distinct polar networks. play a role in proton transduction.) and the discussion (...are likely also involved in mediating proton transduction.) should be clearly presented as speculation until they are validated experimentally.The strongest aspect of this work (aside from the structure itself) is the analysis of TBZ binding. I will comment on some minor points below, but there is one problematic aspect to this analysis. The discussion on how TBZ stabilizes the occluded conformation of VMAT2 is premature without structures of apo-VMAT2 and possibly structures with other ligands bound. We don't really know at this point whether VMAT2 might be in the same occluded conformation in the absence of TBZ. Any statements regarding the effect of interactions between VMAT2 and TBZ depend on demonstrating that TBZ has a conformational effect. The same applies to the discussion of the role of W318 on conformation and to the loops proposed to "occlude the luminal side of the transporter" (line 131).The description of VMAT2 mechanism makes many assumptions that are based on studies with other MFS transporters. Rather than stating these assumptions as fact (VMAT2 functions by alternating access...), it would be preferable to explain why a reader should believe these assumptions. In general, this discussion presents conclusions as established facts rather than proposals that need to be tested experimentally.The MD simulations are not described well enough for a general reader. What is the significance of the different runs? ED Fig. 4d is not high enough resolution to see the details.
**Reviewer #3 (Public Review):**
Summary:The vesicular monoamine transporter is a key component in neuronal signaling and is implicated in diseases such as Parkinson's. Understanding of monoamine processing and our ability to target that process therapeutically has been to date provided by structural modeling and extensive biochemical studies. However, structural data is required to establish these findings more firmly.Strengths:Dalton et al resolved a structure of VMAT2 in the presence of an important inhibitor, tetrabenazine, with the protein in detergent micelles, using cryo-EM and with the aid of protein domains fused to its N- and C-terminal ends, including one fluorescent protein that facilitated protein screening and purification. The resolution of the maps allows clear assignment of the amino acids in the core of the protein. The structure is in good agreement with a wealth of experimental and structural prediction data, and provides important insights into the binding site for tetrabenazine and selectivity relative to analogous compounds. The authors provide additional biochemical analyses that further support their findings. The comparison with AlphaFold models is enlightening.

We appreciate this summary and thank reviewer #3 for their helpful suggestions to improve the manuscript.

Weaknesses:The authors follow up their structures with molecular dynamics simulations of the tetrabenazine-bound state, and test several protonation states of acidic residues in the binding pocket, but not all possible combinations; thus, it is not clear the extent to which tetrabenazine rearrangements observed in these simulations are meaningful. Additional simulations of the substrate dopamine docked into this structure were also carried out, although it is unclear whether this "dead-end" occluded state is a relevant state for dopamine binding. The authors report release of dopamine during these simulations, but it is notable that this only occurs when all four acidic binding site residues were protonated and when an enhanced sampling approach was applied.

As an occluded neurotransmitter bound structure has yet to be solved experimentally, it is not possible to address whether this state resembles the docked dopamine structure. However, it is reasonable to hypothesize that this is a relevant state for dopamine binding and if so, these simulations would be of great interest. The MD simulations which were performed are logical, based on the calculated pKa of the residues and the known pH of the vesicle lumen (5.5). Note that we have carried out a total of more than 2 microseconds of simulations, which required a significant computing time/memory allocation for the current runs in explicit water and membrane. To investigate all possible combinations, it would require at least 16 independent simulations, to be performed in duplicates, to vary protonation status of the four highlighted acidic residues alone, not including proper experimental replicates. We do not believe this to be a feasible suggestion, nor necessary given that the selected combinations were based on rational evaluation of on-path amino acids that were assessed to be potentially protonated.